# A Review on the Development of Gold and Silver Nanoparticles-Based Biosensor as a Detection Strategy of Emerging and Pathogenic RNA Virus

**DOI:** 10.3390/s21155114

**Published:** 2021-07-28

**Authors:** Nadiah Ibrahim, Nur Diyana Jamaluddin, Ling Ling Tan, Nurul Yuziana Mohd Yusof

**Affiliations:** 1Southeast Asia Disaster Prevention Research Initiative (SEADPRI), Institute for Environment and Development (LESTARI), Universiti Kebangsaan Malaysia, Bangi 43600, Selangor, Malaysia; nadiahibrahim35@gmail.com (N.I.); diyanajamaluddin@gmail.com (N.D.J.); 2Department of Earth Sciences and Environment, Faculty of Science and Technology, Universiti Kebangsaan Malaysia, Bangi 43600, Selangor, Malaysia; yuziana@ukm.edu.my

**Keywords:** biosensor, nanoparticles, plasmonic, electrochemical biosensing, nucleic acid, viral disease

## Abstract

The emergence of highly pathogenic and deadly human coronaviruses, namely SARS-CoV and MERS-CoV within the past two decades and currently SARS-CoV-2, have resulted in millions of human death across the world. In addition, other human viral diseases, such as mosquito borne-viral diseases and blood-borne viruses, also contribute to a higher risk of death in severe cases. To date, there is no specific drug or medicine available to cure these human viral diseases. Therefore, the early and rapid detection without compromising the test accuracy is required in order to provide a suitable treatment for the containment of the diseases. Recently, nanomaterials-based biosensors have attracted enormous interest due to their biological activities and unique sensing properties, which enable the detection of analytes such as nucleic acid (DNA or RNA), aptamers, and proteins in clinical samples. In addition, the advances of nanotechnologies also enable the development of miniaturized detection systems for point-of-care (POC) biosensors, which could be a new strategy for detecting human viral diseases. The detection of virus-specific genes by using single-stranded DNA (ssDNA) probes has become a particular interest due to their higher sensitivity and specificity compared to immunological methods based on antibody or antigen for early diagnosis of viral infection. Hence, this review has been developed to provide an overview of the current development of nanoparticles-based biosensors that target pathogenic RNA viruses, toward a robust and effective detection strategy of the existing or newly emerging human viral diseases such as SARS-CoV-2. This review emphasizes the nanoparticles-based biosensors developed using noble metals such as gold (Au) and silver (Ag) by virtue of their powerful characteristics as a signal amplifier or enhancer in the detection of nucleic acid. In addition, this review provides a broad knowledge with respect to several analytical methods involved in the development of nanoparticles-based biosensors for the detection of viral nucleic acid using both optical and electrochemical techniques.

## 1. Introduction

At the end of February 2003, a viral respiratory disease has been reported caused by a coronavirus named severe acute respiratory syndrome-associated coronavirus (SARS-CoV) [1]. The viruses were first identified in China and spread to other countries, which marked the first severe and readily transmissible new disease to emerge in the 21st century [2]. A few years later, a close relative of SARS-CoV was identified in Saudi Arabia in 2012, which was named the Middle East respiratory syndrome (MERS)-CoV [3]. The virus was also detected in other parts of the Middle East including Jordan, Qatar, Oman, and the United Arab Emirates [4]. Both SARS-CoV and MERS-CoV and the current outbreak SARS-CoV-2 belong to the genus betacoronavirus out of four generas (alpha, beta, gamma, and delta), which gain much interest due to the capability of the genus to cross animal–human barriers and emerge to become main human pathogens [5]. Meanwhile in 2014, the resurgence of highly infectious viral disease in West Africa caused by Ebola virus (EBOVs) contributed to a high burden of global mortality [6]. The first outbreak of Ebola was reported in 1976, and the outbreak of Ebola virus disease is still ongoing up to the present [7]. The disease was characterized by systemic viral replication, immune suppression, abnormal inflammatory responses, and major fluid and electrolyte losses that can cause multiple organ dysfunction [8].

In the meantime, another re-emergent pathogenic virus that caused mosquito-borne disease known as Zika virus (ZIKV) was reported in Brazil in 2015 in which the first case was identified in humans in 1952 [9]. The disease is related to neurological complications including encephalitis/meningoencephalitis [10]. Another mosquito-borne viral infection that could lead to lethal complications such as severe bleeding and organ impairment in severe cases is caused by dengue virus (DENV) [11]. In recent decades, the incidence of dengue has been reported to rise dramatically around the world with the increase in number of deaths [12]. In addition, the blood-borne diseases caused by viruses, such as human immunodeficiency virus (HIV), hepatitis B virus (HBV), and hepatitis C virus (HCV), have also become a major global public health issue [13]. These viral infections especially HIV have affected many people in Asian countries, such as in India, Indonesia, Myanmar, Nepal, and Thailand [14]. The most recent pandemic that has afflicted the global health-care systems with high reproductive number (R_0_) is SARS-CoV-2, which belongs to the same family as SARS-CoV and MERS-CoV [15]. This novel coronavirus was isolated by Chinese scientists on the 7th of January 2020 and currently has spread rapidly to other countries [16].

The emergence and re-emergence of pathogenic viruses have caused the age-old battle of humans against these viral infectious diseases. In addition, the problem of rapidly mutating viruses, which is common among the emerging viruses, is adding more difficulties to win the battle against pathogenic viruses [17]. Hence, it is crucial in our health-care system to have a rapid, early, accurate, and on-site diagnosis to contain the spread of the emerging viral diseases with the proper treatment services. Generally, the mainstream methods for the detection of viruses are immunological assay and molecular-based assay [18]. Immunological diagnostic techniques based on antigen–antibody reaction using rapid test kits have been extensively utilized for the detection of human viral infection with a higher detection rate. However, it is ineffective for early or active infection, as the production of antibodies might take several weeks up to a few months [19]. Therefore, the molecular-based assay using real-time quantitative reverse transcription-polymerase chain reaction (qRT-PCR) is widely employed owing to its high sensitivity and specificity toward nucleic acid detection, with a detection limit of as low as 10 viral DNA copies [20]. Nevertheless, qRT-PCR requires expensive, sophisticated equipment and trained personnel as well as time-consuming preparation steps, making it unsuitable for point-of-care (POC) testing [21].

Alternatively, the use of biosensors as POC testing including chip-based and paper-based biosensors, which are typically low-cost and user-friendly, has offered tremendous potential for rapid medical diagnosis [22]. A biosensor can be defined as a sensing device or a measurement system designed for the specific detection of an analyte of interest by using biological interactions, and the resulting signal is transformed into a readable form by using a transducer. The components of a typical biosensor consist of biological receptors and transducers [23]. The bioreceptor specifically interacts with the target analyte, and the transducer plays a role in converting this interaction into an electrical signal. Depending on the type of bioreceptor used, biosensors can broadly be classified into four classes, i.e., DNA biosensor, enzyme biosensor, whole-cell biosensor, and phage biosensor. Meanwhile, according to the different types of transducer used, a biosensor can be categorized into electrochemical biosensor, piezoelectric biosensor, calorimetric biosensor, and optical biosensor [24].

The use of nanotechnology in the development of biosensors has enhanced their performance and sensitivity due to the characteristics of nanoparticles in terms of good biocompatibility, broad structure variety, and notable bioimitative characteristics, which can offer numerous biosensing functions and applications [25]. Nanoparticles are tiny chemical substances or materials that are manufactured and used at a very small scale ranging from 1 nm to 100 nm [26]. Several novel characteristics of nanoparticles, such as good chemical reactivity and high strength conductivity, have attracted much attention for usage in the development of electrochemical biosensors [27]. Various kinds of nanoparticles have been studied in the context of biological detection, including carbon nanotubes, silica nanoparticles, quantum dots, polymeric nanoparticles, and metal nanoparticles (MNPs) [25]. Among these nanoparticles, MNPs such as gold nanoparticles (AuNPs) and silver nanoparticles (AgNPs) have been widely used due to their unique physicochemical features, such as the morphological and structural characterization at the nanoscale [28]. These properties enabled the MNPs to be synthesized and modified with various chemical functional groups for conjugation with DNA probes, antibodies, and ligands, which are crucial in the development of biosensors.

In this review, we will be focusing on the fabrication of MNPs-based biosensors incorporating gold (Au) and silver (Ag) nanoparticles due to their unique optoelectronic properties e.g., tunable plasmonic properties and outstanding electrochemical properties [29]. The excellent electrical conductivity and high sensitivity of both Au [30] and Ag [31,32] have been proven by a very low limit of detection (LOD) toward DNA/RNA, antibody/antigens, and enzymes. In addition, Au and Ag possess high stability against oxidation and inertness in chemical reactions compared to other metals [33]. A general overview on the different types of MNPs-based biosensors for pathogenic RNA virus including SARS-CoV-2 as well as optical and electrochemical transducers associated with biosensor analytical detection strategies are discussed in detail. This manuscript is differed from the previously published reviews as it focuses more on the development of Au/Ag NPs-based biosensors to detect the pathogenic RNA virus. In addition, the advantages and limitations for each type of biosensor have been included in this review. This might help other researchers make decisions and consider the development of a more efficient, cost-effective, and novel biosensor. The last section is also provided with the future aspects and recommendations toward a robust and effective detection strategy of the current pathogenic virus, SARS-CoV-2.

## 2. The Optical and Electrochemical Properties of Metal Nanoparticles

The optical properties for AuNPs and AgNPs have become a research focus since they exhibit the most interesting selective absorption in the visible and near-infrared range. Both noble metals have different plasmon resonance absorption bands with below 500 nm for AgNP [34] and 500–600 nm for AuNP [35], respectively. The plasmon resonance of the MNPs indicates the characteristic of fluorescence quenching or enhancement based on the spectral overlap between fluorophores [36]. The fluorescence quenching is more pronounced for AuNPs due to their strongly absorbing labels [37]. Meanwhile, in the case of AgNPs, the fluorescence enhancement can be observed depending on the size and shape of the particles. Ag nanoclusters (AgNCs) have been reported to offer great potential as ultrabright fluorescents and are brighter than Au nanoclusters (AuNCs) [38]. Due to this characteristic, DNA template AgNCs (DNA/Ag NCs) were developed as a fluorophore to detect DNA/RNA [39], in which the operation of the biosensor has been further detailed in this review in Section 4.1.

In terms of electrochemical properties, there is no difference in the overall performance of using either Au/Ag in the modified electrochemical genosensors based on cobalt porphyrin-DNA [40]. Both MNPs are attached to the Au electrode surface in close proximity through binding with a DNA strand. A robust genosensor demonstrated a considerably improved LOD for complementary ssDNA strands by the inclusion of AuNPs on the electrode surface. In order to investigate the performance of AgNPs, the AuNPs have been replaced with AgNPs in the genosensor [41]. The LOD obtained for the AuNP sensor was 3.8 × 10^–18^ M, whilst the AgNPs sensor yielded an LOD at 5.0 × 10^–18^ M. Hence, the change from the use of Au to Ag did not affect the redox behavior of the cobalt porphyrin. AgNPs have not been directly attached to the gold electrode for DNA detection purposes; however, they were exploited for the general improvement of electron transfer at the gold electrode interface. Nevertheless, the silver system has a better DNA economy, as silver-based compounds are much cheaper than Au [42]. Thus, this contributes to the cost-effectiveness and suitability for mass production of highly sensitive DNA sensors. Further optical and electrochemical properties of MNPs were elaborated in the following subsections.

### 2.1. Gold Nanoparticles

The optical properties of gold nanoparticles (AuNPs) depend on several factors including the size, shape, structure, and morphology of the nanoparticles [26,43]. The surface plasmon resonance (SPR) plays a vital role in light scattering and absorption, which causes different color detection of AuNPs. For example, as the size increases, the color of colloidal suspensions of AuNPs changes from red to blue or purple and vice versa [44]. This phenomenon occurs due to the effect of the SPR peak that shifts slightly toward the red region, which causes the absorption of red light. As a result, the blue light is reflected, and a pale blue or purple-colored solution is observed [43]. Different shapes of gold, such as nanospheres, nanorods, nanoshells, nanowires, nanocubes, nanostars, nanotriangles, and nanocones, have different spectral bands. For instance, Au nanospheres usually exhibit only one plasmon band centered at about 520 nm with an intense absorption peak from 500 to 550 nm. Meanwhile, two SPR bands can be observed when the symmetry is reduced from spherical to cylindrical such as nanorods. Two bands appear due to the elongated particles along one axis [35]. For nanoshells, the plasmon properties depend on the shell thickness, in which the SPR peak shifts from 900 to 1000 nm with a shell thickness in the range of 7 to 5 nm. These tunable properties of AuNPs enable the optical detection of nucleic acid through the fluorescence measurement and colorimetric test.

With regard to the electrochemical properties of AuNPs, this noble metal has an excellent electron transfer ability, which plays a crucial role in the detection sensitivity of an electrochemical biosensor. For instance, AuNPs were used to enhance the electron transfer between electrode and redox species of ferri-ferrocyanide [Fe(CN)_6_]^3−/4−^ in a simple electrochemical DNA sensing method [45]. In the approach, the ssDNA was immobilized onto the gold electrode surface, and it was used to capture AuNPs, as DNA can be adsorbed on the AuNPs due to the affinity interaction between Au and nitrogen-containing bases [46]. Hence, the electrons can be transferred through AuNPs to the electrode surface, enabling the measurement of electrochemical response.

AuNPs possess a redox activity, which can either be oxidized or reduced electrochemically for DNA detection. In a study by Rasheed and Sandhyarani [47], a sandwich-type electrochemical genosensor was developed in which the target DNA was captured by the immobilized probe strands (capture probes) on a graphene-modified glassy carbon electrode (GCE). A reporter probe DNA conjugated to AuNPs was used to hybridize to the other half of the target DNA sequence. Then, the DNA hybridization event was detected through the electrochemical oxidation of AuNPs. In addition, AuNPs can also act as a signal amplifier in an electrochemical detection. The unique conducting properties of AuNPs enable a better sensing performance through the conjugation with other redox molecules. For instance, the redox molecules conjugated to AuNPs, such as methylene blue (Mb) [48], ferrocene (Fc) [49], and doxorubicin [50], can magnify the electrochemical signal in the detection of specific target genes. The integration of AuNPs with the redox molecules can significantly enhance the sensitivity of the resulting sensor/biosensor.

### 2.2. Silver Nanoparticles

Among the noble metals that exhibit plasmon resonance, silver (Ag) displays the highest efficiency of localized surface plasmon resonances (LSPR) excitation with a wider wavelength range [51]. In addition, other characteristics of Ag in plasmonic applications are sharper LSPR bands, less dissipative, and higher refractive index compared to Au. These plasmon properties can be effectively regulated by the size and shape of the nanostructure and also the surrounding dielectric [52]. Several kinds of silver nanoparticles (AgNPs) structures of different shapes that are commonly used include nanorods, nanowires, nanoprisms, and nanoflowers. For the asymmetrically shaped AgNPs, such as nanorods and nanowires, there are two plasmon absorbance bands corresponding to the longitudinal and transverse modes with SPR peaks ranging from 300 to 800 nm measured using a UV-vis spectrometer [53]. For silver nanoprisms, the plasmon band depends on the types of coatings or shells that are used to protect the morphology and surface structure of silver nanoprisms from degradation by halide ions, thiols, UV radiation, heat, and acids. For example, the LSPR band of poly-10 (*N*-isopropylacrylamide) (PNIPAM) modified silver nanoprisms was observed with a small red shift from 514 nm to 516–522 nm, and it demonstrated an increase of absorbance intensities in the complete spectral range at high polymer concentration [54]. Meanwhile, the Ag nanoflowers have raised much attention due to the presence of hot spots in sharp tips and nanoparticle junctions over the whole particle surface, which enable the excellent performance in surface-enhanced Raman scattering (SERS) [55]. This flower-like structure has the features of abundant nanogaps and being spiky near the surface, which can induce broadband plasmonic scattering spectra over the whole visible region with the visible range at 350–750 nm [56].

In addition to the excellent optical properties, AgNPs have a very high electrical conductivity. In a study by Ali et al. [57], the incorporation of AgNPs into polyacrylonitrile fibers to produce carbon nanofibers (CNFs) was found to be an effective means of improving the conductive performance of the CNFs. A concentration-dependent influence of AgNPs on the enhancement in the electrical properties of CNFs was observed. Another study by Santos et al. [58] investigated the electrochemical response of AgNPs incorporated in composite graphite for the fabrication of electrochemical biosensor. Three types of composite electrodes were used, i.e., graphite–epoxy composite dispersion (GEC), the incorporation of AgNPs in GEC (AgNPs/GEC), and mixed with polyaniline (AgNPs/PANI/GEC). Based on the electrochemical results, both composite electrodes modified with AgNPs demonstrated a linear increase of redox peak currents, which implied an improvement of the current sensitivity of the AgNPs-modified composite electrodes. Next, AgNPs can also act as a redox biomolecule, which could undergo the oxidation/reduction process. This redox property enabled the analyte detection based on the dissolution of AgNPs, which were applied in the fabrication of biosensor to detect cholesterol [59]. Cholesterol oxidase (ChOx) was used to oxidize cholesterol into cholest-4-en-3-one and H_2_O_2_, in which H_2_O_2_ was responsible to convert AgNPs into Ag^+^ ions, and resulted in a color change from yellow to pinkish/colorless depending on the concentration of H_2_O_2_ that was produced_._

### 2.3. The Gold-Silver Nanoparticles

The optical and electrochemical properties of Ag-Au bimetallic nanoparticles have been widely studied due to the beneficial coupling of both metals [60]. The Ag core-Au shell nanoparticles (Ag@AuNPs) have demonstrated the desired optical properties calculated using the Mie solution [61]. In the study, the optical properties of a set of Ag@AuNPs were dependent on the amount of Au added in the coating procedure i.e., 5%, 15%, and 25% Au atomic feeding ratios. The UV-vis spectra were collected for each sample of AgNPs and Ag@AuNPs prepared with increasing Au content. Only one maximum SPR peak at 410 nm was observed for the sample of Ag and Ag@AuNPs with 5% Au atomic feeding ratio. For 15% and 25% Au atomic feeding ratios, two SPR peaks were observed at different wavelengths, which indicated the increasing amount of Au in the sample being coated on the AgNPs surface.

In another study by Zhao et al. [62], they have demonstrated the determination of chromium species [Cr(III) and (VI)] in environmental samples by using the oxidized Ag-AuNPs. The screen-printed carbon electrodes (SPCEs) were modified with Ag-AuNPs through electrochemical deposition. Then, the Ag-Au-SPCE were further oxidized for the sensitive detection of Cr(III). The results showed that the Ag-Au-SPCE could contribute to the formation and stabilization of oxides on the surface of AuNPs compared to without the addition of Ag-Au-SPE. The low detection limit can be achieved by using the oxidized Ag-Au NPs. The excellent electrochemical properties of bimetallic Ag-AuNPs have been intensely studied due to their synergistic, electronic, and catalytic properties, which differ from those of individual monometallic nanoparticle (Au or Ag). Avinte et al. [63] reported a unique feature of Ag-Au NPs with respect to their efficiency for dopamine oxidation. The electrocatalytic response of Ag-AuNPs was compared with the individual AuNPs and AgNPs deposited in the same condition on the CNT-modified electrode. The oxidation peak of dopamine occurred at lower potential (around −0.1 V) at the Ag-AuNPs electrode, indicating fast electron transfer kinetics, and the peak height for dopamine has significantly improved, which implied the Ag-AuNPs exhibiting much better catalytic activity for the oxidation of dopamine than individual AgNPs or AuNPs. This signifies the remarkable and excellent electrochemical properties of bimetallic Ag-AuNPs. However, up to our knowledge and literature data, there are not any reports available on the detection of pathogenic RNA virus by using bimetallic Ag-AuNPs.

## 3. The Choice of Ligands

Typically, nanoparticles-based biosensors consist of a surface coating or known as ligand to increase the stability and dispersibility [64]. There are several types of ligands or capping agents, which include organic ligands, polymeric stabilizers, inorganic metals, and metal oxide surfaces. These ligands can stabilize the nanoparticles-based biosensor through electrostatic repulsion and steric stabilization [65]. Citrate stabilization is one of the most common and simplest stabilization approaches of electrostatic repulsion that uses the negative charges to stabilize the colloid against Van der Waals attractive forces [64]. However, at high salt concentration, the electrostatic repulsion largely fails to provide sufficient colloidal stability and leads to the aggregation of nanoparticles. Meanwhile, the steric stabilization approach is not sensitive to the change in ionic strength but is affected by molecular size and capping density. The polymer ligands such as polyethylene glycol adsorbs to the nanoparticles to form a physical barrier, which can prevent the aggregation of nanoparticles in the steric stabilization [66].

The stabilization of nanoparticles can also be done by using both electrostatic and steric stabilization approaches or known as the electrosteric effect. This approach is typically accomplished by using a charged polymer coating on the nanoparticles surface. The charged macromolecules will provide extra electrosteric repulsion, which can prevent the aggregation of nanoparticles. For instance, poly(ferrocenylsilanes) (PFS), which consists of ferrocene units with positively or negatively charged side groups, can be used as a reducing agent and electrosteric stabilization for the synthesis of AuNPs [67]. During the synthesis of AuNPs, the ferrocene units in the polymer chain donate electrons to the gold precursor, tetrachloroaurate ion (AuCl_4_^−^), which changes the charge of Au(III) to Au(0). Meanwhile, the oxidized PFS^+^ becomes more positively charged, thereby stabilizing the AuNPs through the electrosteric stabilization mechanism.

The chemical composition of the nanoparticles surface is one of the factors to determine the choice of ligands. For example, thiols have high affinity toward the gold surface, whereas carboxyl and hydroxyl groups possessing strong binding affinity to iron oxide nanoparticles. The thiol-containing biomolecules have high affinityat the AuNPs surface due to the soft character of both sulfur and gold, which can be well described by the hard-soft acidbase (HSAB) principle. Some thiol-containing compounds, such as glutathione (GSH) [68] and cysteamine [69], are often used to stabilize AuNPs. A study by Moaseri et al. [68] demonstrated that the GSH-capped AuNPs remained electrostatistically stabilized and dispersed at pH above 6.

Whereas for AgNPs, the common stabilizing agents include the anionic species, such as halides, carboxylates, or polyoxoanions that impart a negative charge on the surface of AgNPs, and cationic species such as polyethyleneimine (PEI) and chitosan that create highly positive charges on the AgNPs surface [70]. In a study carried out by Imran et al. [71], chitosan was used as the reducing agent as well as stabilizing agent for the synthesis of AgNPs. The choice of ligands used to stabilize AgNPs based on steric hindrance include organic polymers, such as poly(vinyl alcohol) (PVA) [72], poly(vinylidene fluoride) (PVDF) [73], and polyethylene glycol (PEG) [74]. The stability of PEG-coated AgNPs and the uncoated AgNPs was investigated in a study by Mohamad-Kasim et al. [75] based on the measurement of polydispersity index (PdI) and zeta potential (ZP) (mV). The results showed that the PDI value of PEG-coated AgNPs was smaller than the uncoated AgNPs, which demonstrated better size distributions of the AgNPs modified with organic polymer ligands. On the other hand, the measured ZP values showed that the PEG-coated biologically synthesized AgNPs were highly stable compared to uncoated AgNPs. Overall, the PEG-coated nanoparticles are better than the uncoated-PEG in terms of size distribution, morphology, and stability. The different shapes, ligand modifications, and sensing techniques to determine the optical and electrochemical properties for both AuNPs and AgNPs are illustrated in Figure 1.

## 4. The Development of Optical and Electrochemical Metal Nanoparticles-Based Biosensors for the Detection of Viral RNA

### 4.1. Optical Nanobiosensors

Optical biosensors provide an optical signal (e.g., colorimetry, absorption, reflectance, chemiluminescence, or fluorescence) that is generated directly by a bioreceptor and biomarker or through bio-recognition events [76]. The detected signal from the optical biosensor can be performed by using an optical label-based or label-free system. Briefly, a label-based system involves the use of an optical label, such as fluorescent dyes, enzymes, or nanoparticles, which are employed as a means to measure the optical signal from the bio-recognition event [77]. Then, the optical signal is generated by a colorimetrifc, reflectometric, fluorescent, or luminescent transducer. In contrast, the detected optical signal in a label-free mode is generated directly through the interaction of the analyzed material with the transducer. This approach does not require the involvement of a label and a chemical conjugation step to monitor the bio-recognition event [78]. There are several detection techniques in the optical biosensor, such as fluorescence, absorption-based surface plasmon resonance (SPR), reflectance, and colorimetry. The optical properties of metal nanoparticles (MNPs) i.e., AuNPs and AgNPs, play a key role due to the localized surface plasmon with resonance wavelength in the visible region, and they are having the ability to display different colors based on the specific wavelengths that permit the analyte detection by using optical detection via reflectance-, colorimetry-, SPR- and fluorescence-based biosensors [79].

#### 4.1.1. Fluorescence-Based Assay

Fluorescence-based nucleic acid assay is widely used due to its high sensitivity, diversity, and ease of operation [80]. Fluorescence occurs when a valence electron of molecule or atom is excited from its ground state and causes the emission of a photon when the excited electron returns to its original ground state [81]. Ethidium bromide is commonly used as a non-radioactive fluorescent tag for identifying and visualizing DNA/RNA bands in electrophoresis. Ethidium bromide stains nucleic acid in agarose gel electrophoresis by intercalating itself into the spaces between the base pairs of the double helix. However, ethidium bromide is a potent mutagen in humans, it can be toxic at high concentrations, and SYBR Green staining reagent and organic dyes are often used as a substitute to the classical ethidium bromide dye as a label for the detection of nucleic acid. Nonetheless, the fluorescent measurement can be challenging especially when determining low quantities of DNA/RNA samples due to lower quantum yields or extinction coefficients and lack photostability of the fluorescent dyes [82]. In the early 2000s, the commercial applications of nanotechnology started to attract the interest and attention of researchers [83]. Nanomaterials such as AuNPs and AgNPs have been introduced as the new fluorophores, which can be an alternative for the organic dyes due to their excellent optical properties [28].

A study by Cao et al. [39] used the versatile label-free fluorescent molecular beacon based on a DNA-AgNCs probe to detect the human immunodeficiency virus (HIV) gene, hepatitis B virus (HBV) gene, and human T-lymphotropic virus type I (HTLV-I) gene. The probe was tagged with DNA template AgNCs and guanine-rich sequences (GRSs) at two terminals serving as a signal reporter with a loop whose sequence was complementary to the target DNA. The GRSs enhanced the red fluorescence of AgNCs 200-fold when placed in proximity to the DNA/AgNCs [84]. In the study, the structure of the hairpin-shaped probe allowed the proximity of GRSs to the AgNCs and led to the enhancement of AgNCs’ fluorescence intensity. After hybridization with the target DNA, it resulted in opening of the hairpin loop, which kept AgNCs away from GRSs. The decrease in the fluorescence intensity indicated the detection of disease-related genes, which can be measured by using a spectrofluorometer. By using this approach, the LOD values based on the signal-to-noise ratio of the HIV gene, HBV gene, and HTLV-I gene were obtained at 4.4 nM, 6.8 nM, and 8.5 nM, respectively. The application of AgNCs in the detection of nucleic acid has been widely used with various strategies. A current study by Jia et al., 2021 [85] has demonstrated the concept of ternary complexes that consist of two single strands probes with a different spilt fragment of AgNCs scaffold at the end of the sequence and the target RNA. As a result, bright AgNCs with green emission would be produced at the assembled scaffold. The main advantage from the implementation of DNA-AgNCs as a fluorescent probe is that it involves a one-step fluorescence intensity labeling sensing platform, which enables the development of a cost-effective and easy-to-use biosensor. However, several parameters such as DNA sequence, environment, and structural changes on the emission of DNA-AgNCs need to be considered during the biosensor fabrication [86].

Another study that implemented the mechanism of fluorescence quenching was observed through the application of carbon dots (CDs) and iron magnetic nanoparticle-capped Au (Fe_3_O_4_@Au) [87]. In the study, Fe_3_O_4_@Au was used to adsorb the CDs on its surface. Two probes having unique long sequence regions were designed to target HTLV-1. One of the probes was functionalized with CDs (CDs/DNA) through covalent bonding, and the other probe was used to complete the hybridization (see Figure 2 for the detailed operations). The biosensor was developed based upon the photoluminescence properties of CDs owing to their exceptional fluorescence quenching ability (on/off) in the proximity of quenchers [88]. The complex of the CDs-probe emitted a strong fluorescence intensity measured at the emission wavelength of 460 nm using a fluorescence microplate reader, and it was quenched after the adsorption on the Fe_3_O_4_@Au surface. In the presence of the DNA target, the fluorescence emission of CDs was recovered, since the CDs/DNA hybridized to the complementary DNA (cDNA) and formed double-stranded DNA, which does not adsorb on the Fe_3_O_4_@Au surface. Meanwhile, the non-hybridized oligonucleotides were adsorbed on the Fe_3_O_4_@Au surface to omit unspecific binding with the target, and that increased the high accuracy of target DNA detection. This biosensor was also tested with other viral genes such as HBV and HIV to ensure the specificity of the detection using the designated probes for HTLV-1. However, the fluorescence emission recovery of CDs/DNA decreased significantly in the presence of HBV and HIV genes. This showed that no DNA hybridization occurred with the non-complementary DNAs ascribed to the non-recovery of fluorescence emission of the CDs/DNA probe. The LOD measured in the study was below 10 nM with a linear range from 10 to 320 nM. The benefit of using Fe_3_O_4_@Au particles is the strong magnetic responsiveness, which can be observed in the study through the adsorption of CDs in the absence of target RNA. In another study, the use of Fe_3_O_4_ with AuNPs has demonstrated a stable platform for an easier and efficient separation of pathogens from the complex samples [89]. Nevertheless, the design of this biosensor required multiple steps of fluorescence and quenching measurements to validate the presence of target RNA.

#### 4.1.2. Localized Surface Plasmon Resonance

Optical absorption-based surface plasmon resonance (SPR) is a label-free method, which involves a collective electron oscillation of the surface conduction electrons [90]. The SPR phenomenon occurs when polarized light hits the surface of metal (or other conducting materials) at the interface of two media (usually glass and liquid) with a specific angle. The interaction depends on several factors, such as the size and shape of metal nanoparticles and the nature and composition of the dispersion medium. The direct and real-time changes of refractive index at the sensor surface, which is proportionate to the biomolecule concentration, permit the execution of analyte detection. The recognition elements must be immobilized on the sensor surface to detect a change in the surface refractive index upon the biomolecules binding for the measurement of intensity shift. A practical SPR instrument usually combines an optical detector to measure intensity shift, light source, gold film, prism, and a layer enabling ligand immobilization, which is integrated with a fluidics system enabling a flow-through operation [78].

The occurrence of SPR also leads to the generation of well-known thermoplasmonic at the plasmonic resonance frequency with the required use of nanoparticles [91]. Recently, a dual-functional plasmonic biosensor combining the plasmonic photothermal (PPT) effect and localized surface plasmon resonance (LSPR) sensing transduction was developed for the clinical diagnosis of SARS-CoV-2 [92]. In the study, the two-dimensional gold nanoislands (AuNIs) functionalized with thiol cDNA receptors were used to detect a DNA target through nucleic acid hybridization. The thermoplasmonic heat was generated on the same AuNIs chip through the photoexcitation of highly energetic electrons that quickly dissipated and released the thermal energies. This PPT-induced temperature affected the refractive index variation of the surrounding environment. In the presence of target DNA, the LSPR response level with the photothermal unit was higher than those without the photothermal unit and was proportional to the nucleic acid concentrations. The LOD for the detection of RdRp sequence of SARS-CoV-2 was about 0.22 ± 0.08 pM by using this LSPR-based biosensor. The photothermal unit can also be used to avoid a false positive response signal in the LSPR sensor. This can be tested through the detection of the RdRp sequence of SARS-CoV, which was used as a control in the study to determine the specificity of the biosensor. Based on the biosensor selectivity results, the sensor was capable of forming a specific and full complementary sequence of the RNA target (RdRp-SARS-CoV-2). Meanwhile, the inhibited hybridization of two partially matched sequences of RdRp-SARS-CoV was observed (Figure 3). Therefore, this demonstrates the advantage of using the thermoplasmic heat, which is to improve the specificity of hybridization. This dual functional PPT effect and the LSPR sensing transduction on a single AuNI chip is cost-effective and can significantly enhance the sensing stability, sensitivity, and reliability of the biosensor. However, the detection strategy involving the PPT effect of this biosensor is only suitable for the detection of nucleic acid, and not protein or enzyme, as the photothermal effect can literally denature the protein molecules [93].

This work further extends the utility of the photothermal-assisted plasmonic sensing biosensing system with the introduction of the thermoplasmonic-assisted dual-mode transducing (TP-DMT) concept [94]. The TP-DMT involves two types of approaches, which are an amplification-free-based direct viral RNA detection (direct sensing system) and an amplification-based cyclic fluorescence probe cleavage (CFPC) detection. The same design of a AuNIs chip functionalized with the complementary DNA was used in the study as a multifunctional medium with the roles of nanoabsorber, nanoheater, and nanotransducer. The rapid hybridization between the target sequence and the complementary DNA receptor was achieved due to the thermoplasmonic assistant unit, which constructed a uniform two-dimensional photothermal field on the surface of the AuNI sensor. This enables the direct sensing system detection of the target sequence. Next, the mixture of an Endo-IV and AP-site-modified fluorescent probe was used to perform the secondary CFPC detection. The fluorescent probes initially hybridized to the captured target RNA and then the Endo-IV recognized the AP site in the probes to cleave the probes into two short strands. With the assistance of the local PPT, the two strands were dissociated, and this directly “switched-on” the fluorescent gain medium, which amplified the LSPR response (Figure 4). The measured LOD value using CFPC detection was 0.275 ± 0.051 fM. The advantage of the biosensor is the use of two kinds of approaches in one platform, which can confirm the detection of target RNA. However, the CFPC-based detection is more sensitive than the direct sensing system, as the system is capable of providing reliable biosensing results, especially when detecting nucleic acid at low levels (<0.1 pM). Despite the high sensitivity, several factors that need to be taken into account are the expense of extended reaction time and extra reagents for the CFPC-based detection.

#### 4.1.3. Colorimetric

The colorimetric test is a solution- or solid-based assay, which implements the color change based on the physical properties of the nanoparticles, such as sizes, shapes, and the state of aggregation [95]. Colorimetric assays based on the aggregation of AuNPs and AgNPs have attracted much attention in biomedical applications. The analytes or target DNA are detected based on the color change that can be seen by the naked eye due to the effect of aggregation or dispersion form of the nanoparticles [96]. In a study by Kim et al. [97], a colorimetric assay was developed through a disulfide-induced self-assembly on bare AuNPs. Two thiol modified probes at the 3′ or 5′ ends were used to target partial genomic regions (30 bp) of MERS-CoV along with upstream E protein gene (upE) and encoding open reading frames (ORF). The two target regions were based on the WHO recommendation for potential preclinical screening of MERS-CoV. These specific thiolated probes (SH-DNA) hybridized with the target to form complementary dsDNA, which produced disulfide-induced self-assembled products due to the continuous formation of disulfide bond in the presence of MgCl_2_. The extended self-assembled complex can stabilize bare AuNPs against salt-induced aggregation, as the stability of AuNPs probes is supported by a strong covalent bond that formed between thiol and gold (Au-S), which is mediated by the sulfhydryl (-SH) functional group (Figure 5). The color of the resulting solutions can be analyzed with the naked eye and UV-vis spectrophotometer. In the presence of a viral RNA target, no color change was observed due to the formation of disulfide-induced self-assembled complex, which prevent the aggregation of nanoparticles. Whereas in the absence of the target, a disulfide-induced interconnection between the probes unable to shield the surface of nanoparticles from aggregation with any concentration of salt, and it turned the color from red to blue. The sensitivity or measured LOD was 1 pmol μL^−1^, which is about 6 × 10^11^ copies μL^−1^ and requires only 26 PCR cycles to diagnose MERS-CoV.

The concept of a disulfide-induced self-assembly has overcome several issues of former approaches via crosslinking and non-crosslinking aggregation AuNPs. The issues of concern, such as (i) the need to immobilize oligonucleotides on the AuNPs, (ii) tedious and often unsuccessful salt-aging processes, (iii) the heterogeneity effect due to different functionalized AuNPs that are required for different targets, (iv) the problem of DNA loading on the surface of AuNPs that strongly depends on the oligonucleotide base composition (spacer, linker, and overhang should be used), and (v) the optimization of variables influencing the uniform and reliable loading of DNA (i.e., the effects of salt concentration, spacer composition, nanoparticle size, surfactant, etc.), can be overcome by using this new concept [98]. Nonetheless the use of colorimetric assay still needs to be verified with the use of UV-Vis spectrometer.

The colorimetric test can be applied into a paper-based analytical device (PAD). Recently, a fabrication of paper-based colorimetric DNA biosensors to detect MERS-CoV, mycobacterium tuberculosis (MTB), and human papillomavirus (HPV) DNA [99] was developed using pyrrolidinyl peptide nucleic acid (acpcPNA) as a probe. The acpcPNA probe has a single positive charge from the lysine at the C-terminus to induce the aggregation of nanoparticles [100]. The citrate anion-stabilized AgNPs were used as a colorimetric agent in this study. In the absence of an acpcPNA and target DNA, the AgNPs colloid appeared as a yellow-colored suspension, which indicated the well-dispersion form due to the electrostatic repulsion of citrate anion-stabilized AgNPs. Upon the addition of cationic acpcPNA, an electrostatic interaction occurred between acpcPNA and the AgNPs, which changed the color of the solution from yellow to red, indicating the aggregation of AgNPs. On the other hand, when the target DNA was added, the specific interaction between the acpcPNA and cDNA led to the depletion of acpcPNA–AgNPs interaction. The negative charged from the acpcPNA-DNA binding caused the dispersion of AgNPs, which changed the color from red to yellow again (Figure 6). However, if non-complementary target DNA was added, the acpcPNA remained bound to the AgNPs, which caused the aggregation of AgNPs and produced red coloration. The intensity of the color depends on the concentration of DNA and was measured by using digital cameras and office scanners combined with image processing software to carry out color, hue, and/or intensity measurements.

The fabrication of the paper-based colorimetric DNA biosensor was assembled using the principles of origami, which consisted of two layers. A wax printing technique was used to create PADs, and the Whatman Grade 1 filter paper was used to print the wax design. The top layer consists of four colorimetric detection zones and four control zones. Meanwhile, the base layer provides a sample reservoir with four wax-defined channel extending outwards. The top layer was folded over the base layer to construct the three-dimensional origami paper-based device. The sample solution was added onto the sample reservoir and moved outward to the colorimetric detection zones through the channels to examine the color change. The developed paper-based DNA sensor yielded the detection limits of 1.53 nM (MERS-CoV), 1.27 nM (MTB), and 1.03 nM (HPV) by measuring the color change of AgNPs. The main advantage of using this biosensor is that the use of acpcPNA can be an alternative to the DNA or RNA probe due to its chemically and biologically stable molecules, easy synthesis, and efficient hybridization with the cDNA strands. The PADs implementation has gained renewed interest from researchers around the world in the fabrication of biosensors due to their several advantages, such as simple, inexpensive, portable, and disposable devices, which complement the characteristics of a POC biosensor [101]. Moreover, a semi-quantitative analysis of colorimetric assays can be accomplished by using the image processor. Despite the simple technology, the fabrication of the paper-based biosensor needs a long optimization time and requires competencies and expertise [102].

#### 4.1.4. Reflectance

The principle of a reflectometric sensor is that if a light is imparted into a fiber, a very small amount of light is scattered back from every point in the fiber during its transmission. In biochemical sensors, an additional layer is added between the transducer and the recognition elements to reduce non-specific interaction, which is based on the white-light interference of visible radiation reflected on both interfaces of a layer [103]. A study by Jeningsih et al. [104] has demonstrated a sandwich hybridization strategy of DNAs using a serotype-specific and ultrasensitive microspheres-based DNA opto-sensor to detect dengue virus (DENV). The succinimide-functionalized poly(n-butyl acrylate) (poly(nBA-NAS) microspheres were employed as the DNA hybridization platform and the gold-latex (AuNPs-PSA) sphere was employed as an optical amplification label. The one-step sandwich hybridization recognition involved a pair of a DNA probe, i.e., capture probe (pDNA), and AuNP-PSA reporter label that flanked the target DNA (Figure 7). In the presence of the target DNA, a sandwich hybridization of the cDNA-rDNA-AuNP-PSA complex was formed, which changed the color of the DNA sensor from white to violet and attenuated the reflectance signal over the visible wavelength. The optical DNA biosensor showed a linear reflectance response between 1.0 × 10^−21^ M and 1.0 × 10^−12^ M cDNA (R^2^ = 0.9807) and the LOD of 1 × 10^−29^ M. Additionally, the proposed DNA micro-optode exhibited long stability shelf life of 10 days operational duration, and it was capable of being reused for four consecutive DNA tests. The poly(nBA-NAS) DNA biosensor was able to distinguish oligonucleotides with a single base mutation. The use of AuNPs as an optical DNA hybridization indicator has improved the refractive index change of the DNA biosensor and substantially increased the biosensor sensitivity toward the detection of viral RNA. A validation study of a DNA opto-sensor was performed through the measurements of DENV serotype 2 cDNA concentration on the same patient blood, urine, and saliva samples using a DNA opto-sensor versus standard RT-PCR. There was no significant difference between the results acquired via both methods, and the developed sandwich-type optical DNA biosensor is comparable with the standard RT-PCR reference method for the DENV DNA detection. The main advantage of this work is the application of AuNPs’ optical properties for effective signal transformation of biological interaction into a physical signal. Therefore, the optical biosensors are selective and sensitive devices for the detection of very low levels of chemicals and biological substances without using electrical fields [105]. Unfortunately, these methods mostly require a DNA hybridization label for target detection as well as a long assay duration. These strategies are also susceptible to nucleic acid degradation that could render an unexpected error in the actual nucleic acid assay.

### 4.2. Electrochemical Nanobiosensor

The electrochemical biosensor allows a detectable signal from the DNA hybridization event to be amplified, thus enabling a quantitative detection of RNA/DNA. The signal amplification approaches in the electrochemical sensing devices are crucial to increase the sensitivity of nucleic acid detection [106]. For this purpose, advanced materials-based electrodes, such as carbon, mercury, gold, and graphene have been incorporated in the surface modification of the electrodes and tested for nucleic acid detection. The electrode surface serves as the immobilization site for the bio-recognition elements, which include aptamers or DNA probes for the analyte detection. Basically, DNA probes are immobilized on the electrode surface via several techniques, such as adsorption, covalent bonding, and avidin/streptavidin-biotin interaction for specific interaction with its cDNA. The simplest technique of nucleic acid immobilization is through adsorption, which does not involve DNA probes modification and any chemical reagents. DNA probes are immobilized on the electrode surface via electrostatic adsorption between the negatively charged phosphate groups of DNAs and the positive charges covering the surface. For instance, several previous studies have demonstrated the use of chitosan and cationic polymeric films on the electrode surface to adsorb the ssDNA probes [107,108].

However, several environmental factors, such as ionic strength, pH, and temperature can cause the desorption of the DNA probes from the electrode surface. Therefore, a covalent bonding, which has a high binding strategy, is used to immobilize the DNA probes on the electrode surface. The common covalent attachment methods involve the chemisorption and covalent attachment. Normally, the synthesized DNA probe is linked with the group of thiols (SH) or amines (NH_2_) at the end of 3′ or 5′ to covalently bind to the metal surface or specific functional group introduced to the electrode surface [109]. This method enables a high specific attachment of DNA probe onto the electrode surface and can prevent non-specific binding. Another strategy for DNA probes immobilization on the electrode surface involves the complex formation of avidin (or streptavidin)-biotin, which is a non-covalent bond. There are two common strategies for the immobilization of biotinylated DNA probes, which involve avidin/streptavidin-functionalized electrode through carboxyl group activation and the biotin/avidin (streptavidin)/biotin sandwiches technique. In the presence of target DNA, the detection of DNA hybridization can be determined based on the changes of electrochemical response [110].

The electrochemical response can be detected using screen-printed electrodes (SPEs), which include working, counter, and reference electrodes [111]. The working electrode serves as a site for RNA/DNA immobilization in which the oxidation and reduction reactions occur based on the redox molecules. The function of the counter electrode is to complete the electric circuit so that a current can be supplied to the working electrode. While the reference electrode, which has a known and constant potential, provides a reference for the assessment of other measured parameters [112]. The electrode system in the electrochemical biosensor can be designed and miniaturized into a three-electrode system [111]. The generated signal is quantitatively measured through several techniques, such as conductometry (detects the changes in conductivity of solutions), voltammetry (measures the current at varying potential), or impedance (the opposition of a circuit to the current flow). In this review, we will focus on the voltammetry and impedance techniques due to the high sensitivity and specificity of the detection techniques [113,114].

#### 4.2.1. Voltammetric Detection

Voltammetry is a technique that measures the resulting current (i) as a function of applied potential (E). The applied potential is responsible for a change in the concentration of an electroactive species at the working electrode as the redox reaction taking place. When a suitable potential is reached, the analyte current begins to flow [115]. The current is called a cathodic current when a reduction occurs and an anodic current when oxidation occurs. The frequently used voltammetry-based techniques include cyclic voltammetry (CV), differential pulse voltammetry (DPV), and square wave voltammetry (SWV). Both CV and DPV are commonly used as a tool to evaluate the antioxidant activity of redox active compounds [116], while DPV measurement is more sensitive due to the discrimination of Faradaic currents (electron transfer to and from an electrode) [117]. DPV works on applying amplitude potential pulses on a linear ramp potential, and the current is measured immediately before each potential change. This produces the current variation, which is plotted as a function of potential [118]. In a study by Cajigas et al. [119], DPV was used to measure the electrochemical signal from a reporter probe, which involved the nanomaterial or redox-active molecules. A screen-printed gold electrode (SPAuE) or SPCEs modified with 4-aminoazobenzene (4-AABZ) and hierarchica gold nanostructures (SPCE/AABZ/Au) were used to detect the Zika viral RNA in the presence of Ruthenium (Ru^3+^) as a reporter. The Ru^3+^ was adsorbed into the groove of the DNA sequence to form an electroactive complex through the electrostatic interactions with the DNA phosphate backbone. This complex enabled the amplification of the guanine oxidation signal through the electron transfer between guanine and the electrode [110]. In the presence of the target RNA, it was sandwiched in between the capture probe and the signal probe from the nanobioconjugate. Then, the Ru^3+^ complex was coupled by electrostatic interaction, serving as a reporter of the electrochemical signal measured by DPV. If the target is absent, the resultant DPV signal is much smaller due to no assembly of the genosensor. The measured LOD by using the nanogenosensor was 0.2 fM and 33 fM at the SPAuE and SPCE/Au, respectively. On the other hand, the voltammetry-based biosensor is inherently ultrasensitive toward the determination of ultralow nucleic acid concentration at fM levels [106]. The signal amplification from the nanobioconjugates generated a high signal even at extremely low concentrations of RNA. The high sensitivity of the resulting electrochemical genosensor was attributed to the optimization of several parameters involved, such as enzyme concentration, AuNPs, and ssDNA concentrations, hybridization temperature, and spacer/signal molar ratio. Other factors that must also be considered are salt concentration, pH value, and temperature, which are responsible for the stability of the nanobioconjugates. This would absolutely be needing extra time, cost, and sample handling procedure.

Another homemade circuit based on the voltammetric detection was fabricated by using a simple paper-based approach to rapidly detect the presence of the SARS-CoV-2 N gene from an infected RNA sample [120]. This quantitative paper-based electrochemical sensor chip permitted the digital detection of SARS-CoV-2 genetic material, which is low-cost and easy to implement. The biosensor uses AuNPs capped with highly specific antisense oligonucleotides (ssDNA) targeting viral nucleocapsid phosphoprotein (N-gene). The sensing probes were immobilized on a paper-based electrochemical platform for the fabrication technology of a simple hand-held reader to detect the nucleic acid. In the study, two separate regions of the N-gene were used as target viral RNA using four different thiol-functionalized AuNPs. Two antisense probes, i.e., P1 and P3, were modified with thiol-functionalized at 5’ends, and another two probes, P2 and P4, were modified at 3’ends. There were two different conjugations of ssDNA probes used in this study. The first one was conjugated directly to the gold electrode surface, and for the second configuration, the probes were capped to the surface of AuNPs and then deposited at the electrode surface (Figure 8).

The sensitivity of these two configurations were determined by measuring the output voltage. The higher signal response was observed in the second type of sensor configuration due to the properties of AuNPs, which favored the electron transfer kinetics and also having a large surface area for the binding of ssDNA probes with the target viral RNA, eventually leading to signal amplification. Hence, these ssDNA probes were utilized to cap AuNPs in order to accelerate the electron transfer and loaded over the gold electrode, which had the graphene-coated filter paper for enhancement in the signal of an electrochemical sensor toward the selective detection of SARS-CoV-2 RNA. Then, the successful surface capping of AuNPs with ssDNA probes was confirmed from their relative change in the UV-vis absorbance spectrum. In the presence of the target viral RNA, the DNA hybridization between the probes-capped AuNPs and the target gene sequence occurred on the surface of the graphene film, which caused the increase of charges and induced a positive drift in a variation of the graphene potential. The LOD by using the device was 6.9 copies μL^−1^ measured by using the ratio of the relative change in the sensor voltage (ΔV/V0) to SARS-CoV-2 RNA concentration. The sensor response was validated against the negative controls, e.g., SARS-CoV and MERS-CoV RNAs, which showed no significant change in the output signal compared to SARS-CoV-2. The superiority of this homemade biosensor is its rapid detection within less than 5 min of incubation time. In addition, the use of the latest technology integrated with a microcontroller and an algorithm for the computer interface has enabled the real-time recording of electrical signals, which allowed the sensitive detection of RNA virus SARS-CoV-2. In addition, there is no need for additional redox medium for electron transfer reaction, fast response time to achieve equilibrium response stage, excellence shelf life, and its plausible economic production. However, the threshold value that was used to determine a positive or negative sample may be subject to change with an increasing number of tested clinical samples. Hence, it is crucial for the researchers to set a new threshold value for different numbers of samples.

#### 4.2.2. Impedimetric Detection

Electrochemical impedance spectroscopy (EIS) is a sensitive technique for the analysis of the interfacial properties related to bio-recognition events, e.g., DNA hybridization. The principle of impedance involves the opposition to the current flow in an electrical circuit or known as electron transfer resistance. This technique is useful to monitor changes in electrical properties arising from bio-recognition events at the surfaces of modified electrodes and enables a label-free detection of analytes without using any reporter or specific enzymes [121]. The most commonly used equivalent circuit model is the Randles–Ershler, comprising solution resistance (Rs), charge-transfer resistance (Rct), double-layer capacitance (Cdl) at the electrode surface, and Warburg impedance (W), which is demonstrated as a linear tail at the end of the Nyquist arc [122]. The impedimetric detection can be categorized into faradaic and non-faradaic impedance measurements based on the presence and absence of a redox probe, respectively. The faradaic impedance measures the change in the charge transfer resistance due to the bio-molecular interaction or by the electrostatic repulsion between the target molecules and the electroactive species in the supporting electrolytes solution [123]. For example, [Fe(CN)_6_]^3−/4−^ in the solution are repelled by the negatively charged species, which can increase the electron transfer resistance. The increment in electron transfer resistance can be measured as a function of analyte concentration by using electrochemical impedance spectroscopy, which provides quantitative parameters of electrochemical processes [114].

This faradaic impedance was applied in the electrochemical detection of Ebola virus using the gold SPE [124]. In the study, a thiolated DNA capture probe sequence was immobilized on the SPE surface to allow the hybridization with biotinylated target strand DNA. Later, a biotin–streptavidin conjugation bond was formed through the interaction of biotinylated target strand DNA with streptavidin alkaline phosphatase enzyme, which enabled the electrochemical detection by DPV in the presence of 4-aminophenyl phosphate solution [125]. DPV detection is required to examine the reproducibility of impedance measurement in the DNA hybridization sensing. The redox probe [Fe(CN)_6_]^3−/4−^ was used to measure the R_ct_ values after the DNA hybridization reaction. The interaction of [Fe(CN)_6_]^3−/4−^ with dsDNA formed an electrostatic repulsion due to the presence of negatively charged phosphate backbone of the DNA molecule. As a result, the electron transfer could be blocked, which led to the increase of R_ct_ values (Figure 9). By using this impedance measurement on SPE, the measured LOD was 4.7 nM complementary oligonucleotides. The main benefits of this impedance-based biosensor are that it is cost-effective and enables a simple fabrication of the biosensor, as it is a label-free approach in DNA detection. This will avoid the non-specific signal from the fluorescent probes that can be detected during the DNA hybridization process. Hence, the specificity of RNA detection could be increased. This fabrication also enables the detection of an oxidation signal from the enzymatic product by the DPV technique in the presence of target RNA. Despite the simple design, an issue that needs to be addressed is the non-specific adsorption during impedance biosensing, which can still be solved by using several solutions [126].

A schematic diagram is illustrated in Figure 10 that summarizes the development of the AuNPs- or AgNPs-based biosensors that were described in this review. The low LOD values of viral RNA detection acquired by using various types of Au and Ag nanoparticles-based biosensors are tabulated in Table 1. Based on the table, all the biosensors demonstrated high sensitivity with a low LOD in the range of fM to nM. The lowest LOD can be observed by the reflectance-based biosensor, which showed the LOD value of 1 × 10^−29^ M via a sandwich recognition procedure [104]. However, the biosensor was susceptible to nucleic acid degradation, resulting in unexpected errors in actual nucleic acid detection. Hence, several optimizations and modifications of the biosensors are needed to overcome the problem. Another biosensor that demonstrated the lowest LOD value (0.2 fM) was detected by DPV based on the sandwich hybridization assay [119]. This shows that the sandwich-type biosensor design yields the highest sensitivity, which can afford detecting the target nucleic acid at extremely low concentrations.

## 5. Conclusions

The application of nanotechnology in the biosensor has contributed to the increase of sensitivity and specificity of the viral nucleic acid detection. As is well known, the sensitivity of a biosensor is dependent on the capability of the biosensor to detect the minimum amount of analyte, which is known as the LOD [127]. This is to avoid the false negative results during the early or later stage of viral infection in which the viral load is too low. The AuNPs and AgNPs are excellent candidates for signal amplifiers or enhancers in the optical biosensors due to their distinctive sensing properties [128]. In addition, the evolution of the optical absorptive effects exhibited by plasmonic nanoparticles enable the improvement in the biological sensing. One of the approaches is by using a chaotic signal circuit, which can exhibit powerful nonlinear optical responses arising from chaotic behavior in the areas of photonics and plasmonics [129]. By using this approach, the small changes in plasmonic signals of the nanoparticles can be recorded with the assistance of near-resonance optical excitations.

Nonetheless, the electrochemical DNA biosensors have gained much attention recently due to their fast response time, user-friendliness, high specificity, and high sensitivity [130]. The incorporation of AuNPs and AgNPs in the electrochemical biosensing design can provide a promising novel approach for the construction of the biosensor due to their powerful characteristics such as, highly promoting electron transfer reactions, large surface area, electrical conductivity, good chemical stability, and mechanical robustness. In brief, the electrochemical sensors have been proven to be very simple, cost-effective devices that can be designed into a hand-held device and still retain their high sensitivity in the detection of analyte.

## 6. Recommendations

The intelligent use of nanomaterials is anticipated to enhance the analytical performance of biomolecular electronic devices with high sensitivities and detection limits, especially in the molecular diagnosis of viral infections and currently focusing on SARS-CoV-2. Although limited studies were carried out for SARS-CoV-2 RNA detection, the use of MNPs-based biosensors especially AuNPs and AgNPs have been reported in the detection of several pathogenic viral RNAs, which could pave the way toward efficient and promising biosensing systems for the detection of SARS-CoV-2 RNA. In this context, since the nanomaterials-based biosensors have already shown their high potential toward the diagnosis of numerous viral infections, they may perhaps fulfill the current demand for early, rapid, and accurate diagnosis of the current pandemic, i.e., SARS-CoV-2 cases as well.

## Figures and Tables

**Figure 1 sensors-21-05114-f001:**
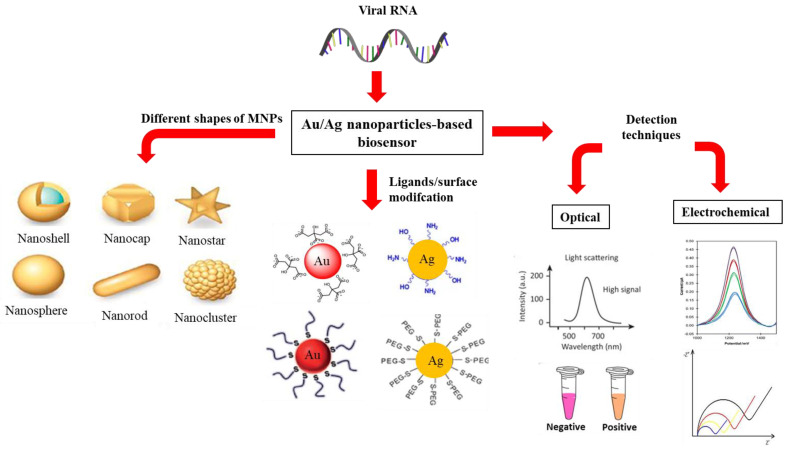
The schematic figure that represents the different shapes of MNPs (AuNPs or AgNPs) and the use of ligands to stabilize the MNPs. The optical and electrochemical properties of the MNPs have been validated by using the optical and electrochemical sensing techniques.

**Figure 2 sensors-21-05114-f002:**
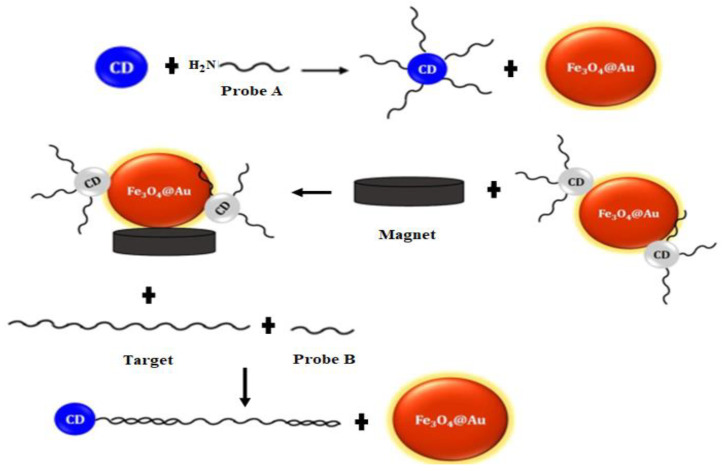
Schematic illustration of the detection steps of DNA target for the HTLV-1 gene based on the application of carbon dots (CDs) and iron magnetic nanoparticle-capped Au (Fe_3_O_4_@Au). Both probes A and B were used to hybridize with the target DNA sequence. Reproduced from Zarei-Ghobadi et al. [87].

**Figure 3 sensors-21-05114-f003:**
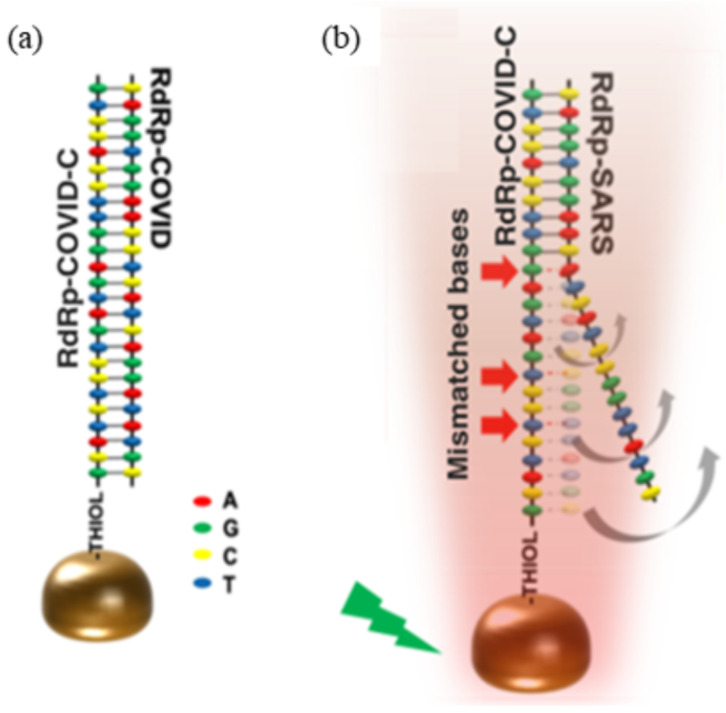
The diagram represents the formation of (**a**) complementary sequences of RdRp-COVID and (**b**) the partially matched sequences of RdRp-SARS. Reproduced from Qiu et al. [92].

**Figure 4 sensors-21-05114-f004:**
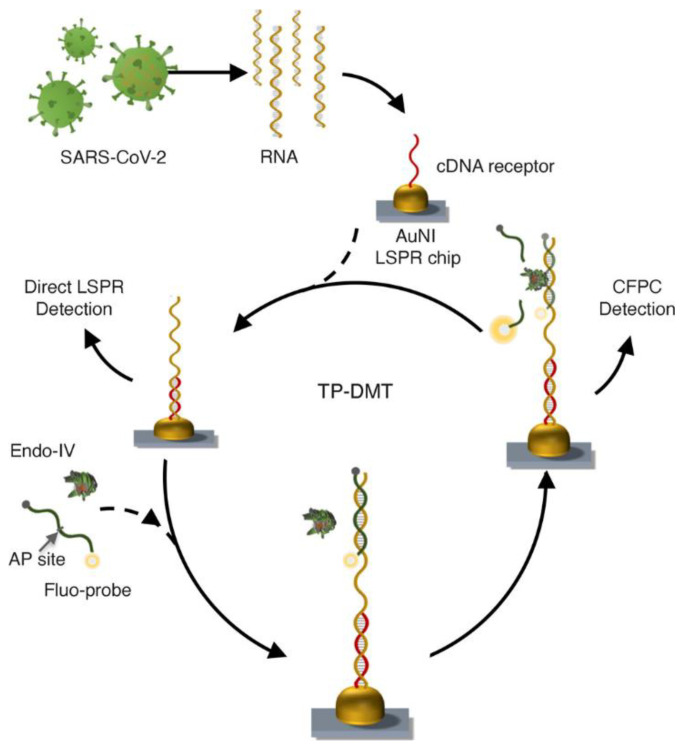
Schematic of TP-DMT viral sensing workflow with two types of approaches, which are the amplification-free-based direct viral RNA detection (direct sensing system) and the amplification-based cyclic fluorescence probe cleavage (CFPC) detection. Reproduced from Qiu et al. [94].

**Figure 5 sensors-21-05114-f005:**
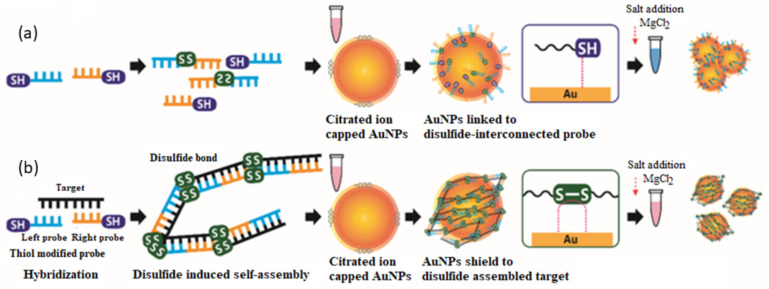
The schematic diagram demonstrated the formation of (**a**) a disulfide-induced interconnection between the AuNPs in the absence of target DNA and (**b**) disulfide-induced self-assembly in the presence of target DNA with the addition of MgCl_2_. Reproduced from Kim et al. [97].

**Figure 6 sensors-21-05114-f006:**
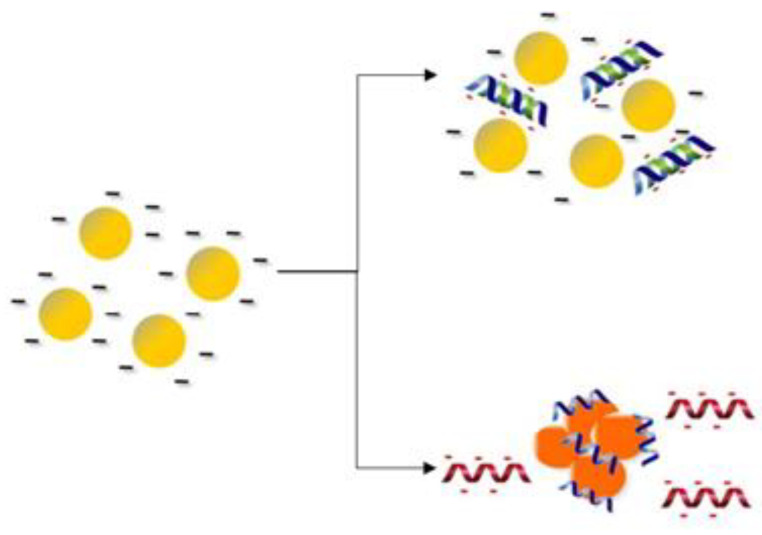
The schematic diagram shows a colorimetric detection that involves the color change of AgNPs. In the presence of target DNA, the AgNPs solution remains yellow (non-aggregated), and in the absence of target DNA, the solution changes to red. Reproduced from Teengam et al. [99].

**Figure 7 sensors-21-05114-f007:**
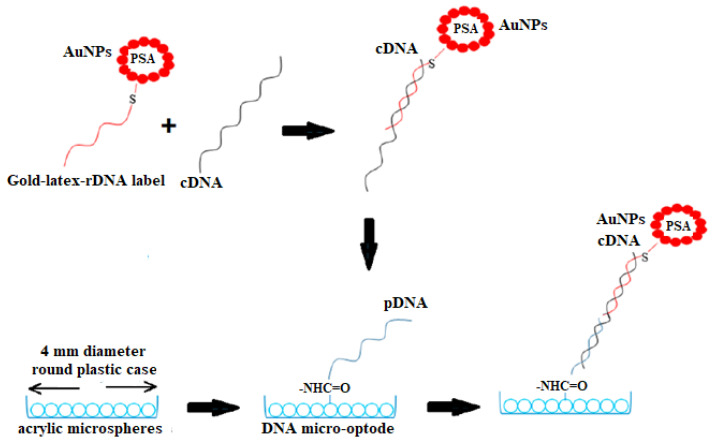
Schematic diagram showing the one-step sandwich hybridization recognition procedure, which involves a DNA capture probe (pDNA) immobilized on the acrylic microspheres, and target (cDNA) and reporter (rDNA) probes labeled with gold nanoparticle–poly(styrene-*co*-acrylic acid) latex (AuNP–PSA) spheres. Reproduced from Jeningsih et al. [104].

**Figure 8 sensors-21-05114-f008:**
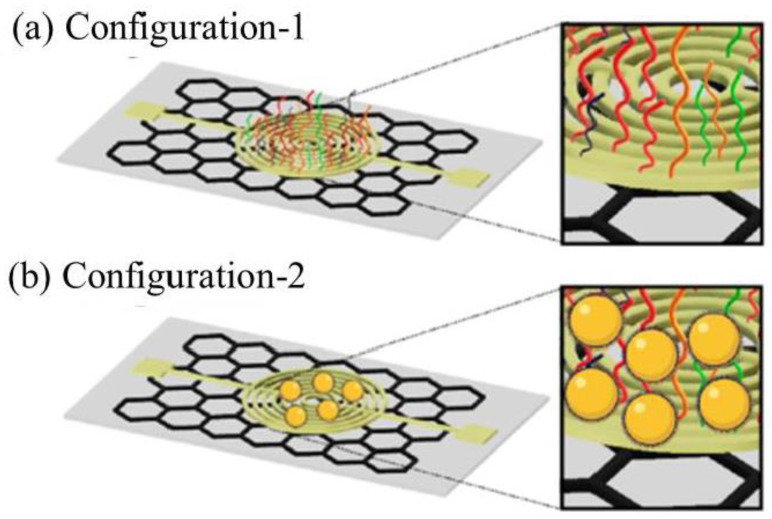
The types of configurations, which involved the (**a**) direct conjugation of probes to the gold electrode surface and (**b**) the probes were conjugated to the AuNPs and then deposited at the sensor platform. Reproduced from Alafeef et al. [120].

**Figure 9 sensors-21-05114-f009:**
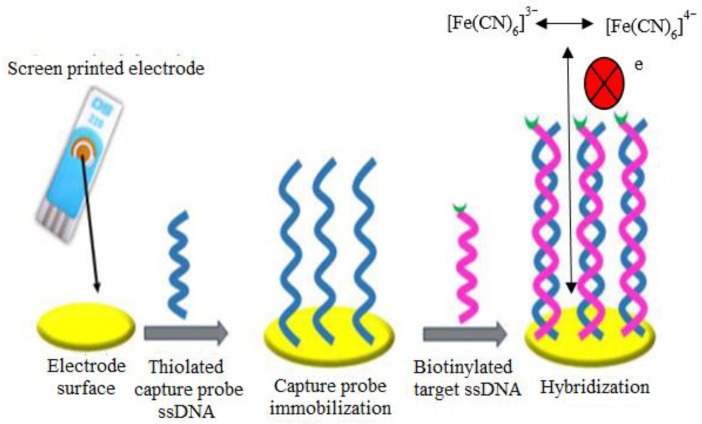
The illustration diagram represents the detection of target DNA by using a capture probe (SH-DNA) immobilized onto the SPE surface. In the presence of target DNA, the electron transfer is blocked due to the electrostatic repulsion that occurs between the interaction of [Fe(CN)_6_]^3−/4−^ with dsDNA. Reproduced from Ilkhani and Farhad [124].

**Figure 10 sensors-21-05114-f010:**
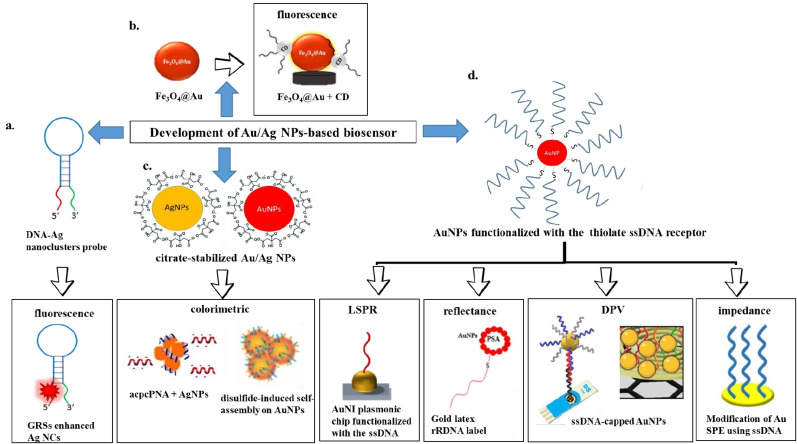
The schematic diagram that summarizes the development of AuNPs- or AgNPs-based biosensors.

**Table 1 sensors-21-05114-t001:** A summary of the different types of Au and Ag nanoparticles-based biosensors used in the detection of pathogenic RNA viruses. The literature citation is indicated under the column of LOD.

Nanoparticles	Design of the Probe	The Approaches for Analyte Detection	Target Genes	Technique	LOD
AgNCs	The probe is tagged with AgNCs and GRSs two terminals with a loop whose sequence was complementary to cDNA (HIV, HBV and HTLV-I)	The detection of analyte was based on the fluorescence quenching of AgNCs in the presence of cDNA, which open the hairpin-shape probe and keeps AgNCs away from GRSs. This results in the decrease of fluorescence intensity.	HIV gene, HBV gene and HTLV-I gene	Fluorescence	HIV = 4.4 nM HBV = 6.8 nM HTLV-I = 8.5 nM [39]
CDs and Fe_3_O_4_@Au	Two probes were designed; CDs/DNA (probe A) and probe (B) to complete the DNA hybridization	The detection of analyte was based on the fluorescence quenching of CDs in the proximity of Fe_3_O_4_@Au. In the presence of cDNA, the fluorescence emission of CDs was recovered, since the CDs/DNA hybridized with the cDNA and formed double-strand DNA, which cannot adsorb on the Fe_3_O_4_@Au surface	HTLV-I	Fluorescence	10 nM [87]
The two-dimensional AuNIs	The AuNI surface was functionalized with thiol-cDNA	A dual-functional plasmonic biosensor combining the PPT effect and LSPR sensing transduction was used to detect the analyte.	SARS-CoV-2	LSPR	0.22 ± 0.08 pM [92]
		The LSPR transduction signal based on the hybridization between the target sequences and the functionalized thiol-DNA receptors was involved in the direct sensing system. In CFPC-based detection, the mixture of API-site-modified fluorescent probe and endo IV were used. The cleaved strand of the probes stimulates the LSPR response.	SARS-CoV-2	LSPR	Direct sensing system = 0.1 ± 0.04 pM CFPC-based detection = 0.275 ± 0.051 fM [94]
AuNPs	Two thiol modified probes at the 5′ site (right) and 3′ site (left)	The formation of a disulfide-induced interconnection between the probes in the absence of target caused AuNPs aggregation (color changed from red to blue) and the formation of disulfide-induced long self-assembled complex prevented AuNPs aggregation in the presence of target (no color change).	MERS-CoV	Colorimetric test	1 pmol μL^−1^ [97]
AgNPs	The acpcPNA was designed as a probe.	The color change of AgNPs (yellow to red) was observed due to the electrostatic interaction between acpcPNA with AgNPs (aggregated). In the presence of cDNA, the interaction of acpcPNA and DNA led to the depletion of acpcPNA-AgNPs interaction, which changed the color from red to yellow again (non-aggregated).	MERS-CoV MTB HPV	Colorimetric test	MERS-CoV: 1.53 nM MTB: 1.27 nM HPV: 1.03 nM [99]
AuNPs-PSA (latex)	AuNPs-PSA latex particles were attached to the thiolated reporter probe (rDNA) by Au–thiol.	Sandwich hybridization strategy of DNAs was performed on poly(nBA-NAS) microspheres. Microsphere poly(nBA-NAS) conjugated to the aminated pDNA via a peptide covalent bond hybridized to the cDNA-rDNA-AuNP-PSA complex, which attenuated the reflectance signal.	DENV serotype 2	Reflectance spectrophotometer	1 × 10^−29^ M [104]
SPAuE or a SPCE/Au	ssDNA monolayer consisting of thiolated signal and spacer probes linked to AuNPs	In the presence of the target RNA, it is sandwiched in between the capture probe and the signal probe from the nanobioconjugate. Then, the Ru^3+^ complex is coupled by electrostatic interaction, serving as a reporter of the electrochemical measured by DPV. If the target is absent, the resultant DPV signal is much smaller due to no assembly of the genosensor.	Zika virus	DPV	SPAuE: 0.2 fM SPCE/Au: 33 fM [119]
AuNPs	Two antisense probes, P1 and P3, were modified with thiol at the 5′ end and another two thiol-modified antisense probes at the 3′ end (P2 and P4)	In the presence of SARS-CoV-2 RNA, the specific RNA−DNA hybridization led to the change in charge and electron mobility on the graphene surface, which brought the change in sensor output voltage.	SARS-CoV-2	DPV	6.9 copies μL^−1^ [120]
AuNPs modified SPE	A thiolated DNA capture probe sequence was immobilized on the SPE surface	The redox probe [Fe(CN)_6_]^3−/4−^ was used to measure the R_ct_ values after the DNA hybridization. The interaction of [Fe(CN)_6_]^3−/4−^ with dsDNA formed an electrostatic repulsion due to the presence of negatively charged phosphate backbone of DNA molecule. As a result, the electron transfer could be blocked, which led to the increase of impedance value (R_ct_ values).	Ebola virus	EIS	4.7 nM [124]

## Data Availability

No new data were created or analyzed in this study. Data sharing is not applicable to this article.

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
