# Peer review of "A Review on the Development of Gold and Silver Nanoparticles-Based Biosensor as a Detection Strategy of Emerging and Pathogenic RNA Virus"

_sensors, 2021, doi:10.3390/s21155114_

Round 1
Reviewer 1 Report
The paper reports an overview of MNPs-based biosensors incorporating with gold (Au) and silver (Ag) nanoparticles for pathogenic RNA virus. This work is well-conducted, the current version, however, fails in view of few different aspects: 1. In the section 2. The optical and electrochemical properties of metal nanoparticles, the reason why to choose gold (Au) and silver (Ag) nanoparticles for this review, and the difference in the optical and electrochemical properties of these two nanoparticles, the comparison is needed. And what about gold- silver nanoparticles? 2. The schematic figure is relatively lacking, especially in the Section 2 and 3. Besides, two or more pictures can be combined into one figure. 3. Section 3, the choice of ligands, it would be more clear if it was wrote in terms of the principles of stabilization. 4. The overlook seems too narrow for the readers.Author Response
Response to Reviewer #1
The paper reports an overview of MNPs-based biosensors incorporating with gold (Au) and silver (Ag) nanoparticles for pathogenic RNA virus. This work is well-conducted, the current version, however, fails in view of few different aspects:
- In the section 2. The optical and electrochemical properties of metal nanoparticles, the reason why to choose gold (Au) and silver (Ag) nanoparticles for this review, and the difference in the optical and electrochemical properties of these two nanoparticles, the comparison is needed. And what about gold- silver nanoparticles?
ANSWER- The excellent electrical conductivity and high sensitivity of both Au [30] and Ag [31,32] have been proven by a very low limit of detection (LOD) towards DNA/RNA, antibody/antigen and enzymes. Besides, Au and Ag possess high stability against oxidation and inertness in chemical reactions compare to other metals [33]. The reason why to choose Au and Ag for this review is added in the introduction. Correction has been made in page 3, line 31-35 of the revised manuscript.
- The optical and electrochemical properties of metal nanoparticles
ANSWER- The optical properties for AuNPs and AgNPs have become a research focus since they exhibit the most interesting selective absorption in the visible and near-infrared range. Both noble metals have different plasmon resonance absorption bands with below 500 nm for AgNP [34] and at 500-600 nm for AuNP [35], respectively. The plasmon resonance of the MNPs indicates the characteristic of fluorescence quenching or enhancement based on the spectral overlap between fluorophores [36]. The fluorescence quenching is more pronounced for AuNPs due to their strongly absorbing labels [37]. Meanwhile in the case of AgNPs, the fluorescence enhancement can be observed depending on the size and shape of the particles. Ag nanoclusters (AgNCs) have been reported to offer great potential as ultrabright fluorescent and are brighter than Au nanoclusters (AuNCs) [38]. Due to this characteristic, DNA template AgNCs (DNA/Ag NCs) were developed as a fluorophore to detect DNA/RNA [39], in which the operation of the biosensor has been further detailed in this review in section 4.1.
In terms of electrochemical properties, there is no difference in the overall performance of using either Au/Ag in the modified electrochemical genosensors based on cobalt porphyrin-DNA [40]. Both MNPs are attached to the Au electrode surface in close proximity through binding with DNA strand. A robust genosensor demonstrated a considerably improved LOD for complementary ssDNA strands by inclusion of AuNPs on the electrode surface. In order to investigate the performance of AgNPs, the AuNPs have been replaced with AgNPs in the genosensor [41]. The LOD obtained for AuNP sensor was 3.8×10–18 M, whilst AgNP sensor yielded a LOD at 5.0×10–18 M. Hence, the change from the use of Au to Ag did not affect the redox behavior of the cobalt porphyrin. AgNPs have not been directly attached to the gold electrode for DNA detection purposes, however they were exploited for general improvement of electron transfer at the gold electrode interface. Nevertheless, the silver system has a better DNA economy as silver-based compounds are much cheaper than Au [42]. Thus, this contributes to the cost-effectiveness and suitable for mass production of highly sensitive DNA sensors. Further optical and electrochemical properties of MNPs were elaborated in the following subsections. The differences in the optical and electrochemical properties of Au and Ag nanoparticles are added in section 2. Correction has been made in page 4, line 1-27 of the revised manuscript.
2.3 The gold-silver nanoparticles
The optical and electrochemical properties of Ag-Au bimetallic nanoparticles have been widely studied due to the beneficial coupling of both metals [60]. The Ag core Au shell nanoparticles (Ag@AuNPs) has demonstrated the desired optical properties calculated using the Mie solution [61]. In the study, the optical properties of a set of Ag@AuNPs were dependent on the amount of Au added in the coating procedure i.e., 5%, 15% and 25% Au atomic feeding ratios. The UV-vis spectra were collected for each sample of AgNPs and Ag@AuNPs prepared with increasing Au content. Only one maximum SPR peak at 410 nm was observed for the sample of Ag and Ag@AuNPs with 5% Au atomic feeding ratio. For 15% and 25% Au atomic feeding ratios, two SPR peaks were observed at different wavelengths, which indicated the increasing amount of Au in the sample being coated on the AgNPs surface.
In another study by Zhao et al. [62], they have demonstrated the determination of chromium species [Cr(III) and (VI)] in environmental samples by using the oxidized Ag-AuNPs. The screen-printed carbon electrodes (SPCEs) were modified with Ag-AuNPs through electrochemical deposition. The Ag-Au-SPCE were then further oxidized for the sensitive detection of Cr(III). The results showed that the Ag-Au-SPCE could contribute to the formation and stabilization of oxides on the surface of AuNPs compared to without the addition Ag, Au-SPE. The low detection limit can be achieved by using the oxidized Ag-Au NPs. The excellent electrochemical properties of bimetallic Ag-AuNPs have been intensely studied due to their synergistic, electronic and catalytic properties, which differ from those of individual monometallic nanoparticle (Au or Ag). Avinte et al. [63] reported a unique feature of Ag-Au NPs with respect to their efficiency for dopamine oxidation. The electrocatalytic response of Ag-AuNPs was compared with the individual AuNPs and AgNPs deposited in the same condition on the CNT-modified electrode. The oxidation peak of dopamine occurred at lower potential (around -0.1 V) at the Ag-AuNPs electrode, indicating fast electron transfer kinetics, and the peak height for dopamine has significantly improved, which implied the Ag-AuNPs exhibiting much better catalytic activity for the oxidation of dopamine than individual AgNPs or AuNPs. This signifies the remarkable and excellent electrochemical properties of bimetallic Ag-AuNPs. However, up to our knowledge and literature data, there are not any reports available on the detection of pathogenic RNA virus by using bimetallic Ag-AuNPs. The explanations on the gold-silver nanoparticles are added into section 2.3. Correction has been made in page 6, line 21-43, and in page 7, line 1-6.
- The schematic figure is relatively lacking, especially in the Section 2 and 3. Besides, two or more pictures can be combined into one figure.
ANSWER- The schematic figure is added in page 8, line 13-32.
- Section 3, the choice of ligands, it would be more clear if it was wrote in terms of the principles of stabilization.
ANSWER- Typically, nanoparticles-based biosensors consist of a surface coating or known as ligand to increase the stability and dispersibility [64]. There are several types of ligands or capping agents, which include organic ligands, polymeric stabilizers, inorganic metals and metal oxide surfaces. These ligands can stabilize the nanoparticles-based biosensor through electrostatic repulsion and steric stabilization [65]. Citrate-stabilization is one of the most common and simplest stabilization approaches of electrostatic repulsion that uses the negative charges to stabilize the colloid against Van der Waals attractive forces [64]. However, at high salt concentration, the electrostatic repulsion largely fails to provide sufficient colloidal stability and leads to the aggregation of nanoparticles. Meanwhile the steric stabilization approach is not sensitive to the change in ionic strength but is affected by molecular size and capping density. The polymer ligands such as polyethylene glycol adsorbs to the nanoparticles to form a physical barrier, which can prevent the aggregation of nanoparticles in the steric stabilization [66].
The stabilization of nanoparticles can also be done by using both electrostatic and steric stabilization approaches or known as the electrosteric effect. This approach is typically accomplished by using a charged polymer coating on the nanoparticles surface. The charged macromolecules will provide extra electrosteric repulsion, which can prevent the aggregation of nanoparticles. For instance, poly(ferrocenylsilanes) (PFS), which consists of ferrocene units with positively or negatively charged side groups can be used as a reducing agent and electrosteric stabilization for the synthesis of AuNPs [67]. During the synthesis of AuNPs, the ferrocene units in the polymer chain donates electrons to the gold precursor, tetrachloroaurate ion (AuCl4¯), which changes the charge of Au(III) to Au(0). Meanwhile, the oxidized PFS+ becomes more positively charged thereby stabilizes the AuNPs through the electrosteric stabilization mechanism.
The chemical composition of the nanoparticles surface is one of the factors to determine the choice of ligands. For example, thiols have high affinity towards the gold surface, whereas carboxyl and hydroxyl groups possessing strong binding affinity to iron oxide nanoparticles. The thiol-containing biomolecules have high affinity for AuNPs surface due to the soft character of both sulfur and gold, which can be well described by the hard-soft acid-base (HSAB) principle. Some thiol‐containing compounds, such as glutathione (GSH) [68] and cysteamine [69] are often used to stabilize AuNPs. A study by Moaseri et al. [68] demonstrated the GSH-capped AuNPs remained electrostatistically stabilized and dispersed at pH above 6.
Whereas for AgNPs, the common stabilizing agents include the anionic species, such as halides, carboxylates or polyoxoanions that impart a negative charge on the surface of AgNPs, and cationic species, such as polyethyleneimine (PEI) and chitosan that create highly positive charges on the AgNPs surface [70]. In a study carried out by Imran et al. [71], chitosan was used as the reducing agent as well as stabilizing agent for the synthesis of AgNPs. The choice of ligands used to stabilize AgNPs based on steric hindrance include organic polymers, such as poly(vinyl alcohol) (PVA) [72], poly(vinylidene fluoride) (PVDF) [73] and polyethylene glycol (PEG) [74]. The stability of PEG-coated AgNPs and the uncoated AgNPs was investigated in a study by Mohamad-Kasim et al. [75] based on the measurement of polydispersity index (PdI) and zeta potential (ZP) (mV). The results showed that the PDI value of PEG-coated AgNPs was smaller than the uncoated AgNPs, which demonstrated better size distributions of the AgNPs modified with organic polymer ligands. The measured ZP values, on the other hand, showed that the PEG-coated biologically synthesized AgNPs were highly stable compared to uncoated AgNPs. Overall, the PEG-coated nanoparticles are better than the uncoated-PEG in terms of size distribution, morphology and stability. The choice of ligands is revised to focus on the principles of stabilization by removing the points that are not related to the principles of stabilization. Correction of the choice of ligands has been made in page 7, line 8-43, and in page 8, line 1-11 of the revised manuscript.
- The overlook seems too narrow for the readers.
ANSWER- Based on the several comments from all reviewers, this review has been edited and added with many additional information (for texts that are marked in red) to give more understanding in the development of gold and silver nanoparticles-based biosensor as a detection strategy of emerging and pathogenic RNA virus. This review also has wrapped-up the advantages and limitations for each biosensor type discussed. This might be helpful for other researchers to make decision and consideration for the development of a more efficient, cost-effective and novel biosensor.

Reviewer 2 Report
The manuscript ID sensors-1310306 corresponds to a review involving particular studies related to Virus Biosensors assisted by Gold and Silver Nanoparticles. Here are some points to the authors:
- It should be clearly stated in the text what this review adds beyond previous reviews on the same topic.
- In my opinion, the keyword noble-metal should be substituted in order to promote the particular findings of this research. You can consider the topic focused in the report as stated in the introduction: Plasmonics or Electrochemical biosensing.
- The inclusion of a graphical roadmap would help to easily visualize the biosensors analyzed in this report.
- The main advantages and disadvantages of the different biosensors studied should be discussed with better details.
- If possible, the authors should distinguish between sensitivity of the different biosensors presented. Do the cited studies address that? Please argue.
- The authors state “the absorption band splits into two parts when the symmetry is reduced from spherical to cylindrical such as nanorods,” However, this statement should be edited, since the optical absorption band in plasmonic nanoparticles always have two components regarding the vectorial nature of light. When symmetry between the minor and the mayor axis in a nanoparticle is present, a coincidence in the Surface Plasmon Resonance bands occurs. Otherwise a shift between the absorption bands can be easily observed by the orthogonal components of polarization of light. You can see for instance: doi:10.1088/2040-8978/14/12/125203. It is suggested to clarify this information within the text.
- A summarizing schematic figure for describing the information in part 4 would be helpful. Maybe a small figure would be nice to summarize the current knowledge.
- The discussion section should present a confrontation of the main results with comparative publication promoting additional solutions in biological sensing by nanoparticles. You can consider for instance nonlinear optic effects or signal modulation. You can see for instance: doi:10.3390/s19214728
- Only few references were included in the bibliography for this year 2021. I consider that some references can be updated in order to see the current panoramic research in this field.
- Collective citations in the style of [41,42] should be individually presented in order to better justify the importance of each citation selected for this review.
Author Response
Response to Reviewer #2
The manuscript ID sensors-1310306 corresponds to a review involving particular studies related to Virus Biosensors assisted by Gold and Silver Nanoparticles. Here are some points to the authors:
- It should be clearly stated in the text what this review adds beyond previous reviews on the same topic.
ANSWER- This manuscript is differed from the previously published reviews as it focuses more on the development of Au/Ag NPs-based biosensor to detect the pathogenic RNA virus. Besides the advantages and limitations for each type of biosensor have been wrapped-up in this review. This might be helpful to other researchers to make decision and consideration for the development of a more efficient, cost-effective and novel biosensor. The difference of this manuscript from the previous reviews is added in the introduction section. Correction has been done in page 3, line 37-42 of the revised manuscript.
- In my opinion, the keyword noble-metal should be substituted in order to promote the particular findings of this research. You can consider the topic focused in the report as stated in the introduction: Plasmonics or Electrochemical biosensing.
ANSWER- The keyword noble-metal has been substituted to plasmonic and electrochemical biosensing in the abstract, in page 1, line 37-38 of the revised manuscript.
- The inclusion of a graphical roadmap would help to easily visualize the biosensors analyzed in this report.
ANSWER- The schematic figure is added in page 8, line 13-32.
- The main advantages and disadvantages of the different biosensors studied should be discussed with better details.
ANSWER- The application of AgNCs in the detection of nucleic acid has been widely used with various strategies. A current study by Jia et al. 2021 [85], has demonstrated the concept of ternary complexes that consist of two single strands probes with a different spilt fragment of AgNCs scaffold at the end of the sequence and the target RNA. As a result, a bright AgNCs with green emission would be produced at the assembled scaffold. The main advantage from the implementation of DNA-AgNCs as a fluorescent probe is that it involves one-step fluorescence intensity labeling sensing platform, which enable the development of cost-effective and easy-to-use biosensor. However, several parameters such as DNA sequence, environment and structural changes on the emission of DNA-AgNCs need to be considered during the biosensor fabrication [86]. The main advantages and disadvantages of the fluorescence-based assay using AgNCs are added in section 4.1.1. Correction has been made in page 10, line 7-15 of the revised manuscript.
The benefit of using Fe3O4@Au particles is the strong magnetic responsiveness, which can be observed in the study through the adsorption of CDs in the absence of target RNA. In another study, the use of Fe3O4 with AuNPs has demonstrated a stable platform for an easier and well separation of pathogens from the complex samples [89]. Nevertheless, the design of this biosensor required multiple steps of fluorescence and quenching measurements to validate the presence of target RNA. The main advantages and disadvantage of the fluorescence-based assay using Fe3O4@Au particles are added in section 4.1.1. Correction has been made in page 10, line 35-40 of the revised manuscript.
Based on the biosensor selectivity results, the sensor was capable to form a specific and full complementary sequence of the RNA target (RdRp-SARS-CoV-2). Meanwhile the inhibited hybridization of two partially matched sequences of RdRp-SARS-CoV was observed (Figure 2). Therefore, this demonstrates the advantage of using the thermoplasmic heat, which is to improve the specificity of hybridization. Besides, this dual functional PPT effect and the LSPR sensing transduction on a single AuNI chip is cost-effective and can significantly enhance the sensing stability, sensitivity, and reliability of the biosensor. However, the detection strategy involving PPT effect of this biosensor is only suitable for the detection of nucleic acid, and not protein or enzyme as the photothermal effect can literally denature the protein molecules [93]. The main advantages and disadvantage of the localized surface plasmon resonance using AuNIs chip are added in section 4.1.2. Correction has been made in page 12, line 7-16 of the revised manuscript.
The advantage of the biosensor is the use of two kinds of approach in one platform, which can confirm the detection of target RNA. However, the CFPC-based detection is more sensitive than the direct sensing system as the system is capable to provide a reliable biosensing results especially when detecting the nucleic acid at low levels (<0.1 pM). Despite of the high sensitivity, several factors that need to be taken into account are such as the expense of extended reaction time and extra reagents for the CFPC-based detection. The main advantages and disadvantage of the localized surface plasmon resonance using AuNIs chip are added in section 4.1.2. Correction has been made in page 13, line 5-10 of the revised manuscript.
The concept of a disulfide-induced self-assembly has overcome several issues of former approaches via crosslinking and non-crosslinking aggregation AuNPs. The concern issues such as i) the need to immobilize oligonucleotides on the AuNPs, ii) tedious and often unsuccessful salt-aging processes, iii) the heterogeneity effect due to different functionalized AuNPs that are required for different targets, iv) the problem of DNA loading on the surface of AuNPs that strongly depends on the oligonucleotide base composition (spacer, linker and overhang should be used) and v) the optimization of variables influencing the uniform and reliable loading of DNA (i.e. the effects of salt concentration, spacer composition, nanoparticle size, surfactant, etc.) can be overcome by using this new concept [98]. Nonetheless the use of colorimetric assay still needs to be verified with the use of UV-Vis spectrometer. The main advantages and disadvantage of the colorimetric assay using a disulfide induced self-assembled products on AuNPs are added in section 4.1.3. Correction has been made in page 14, line 10-19 of the revised manuscript.
The main advantage of using this biosensor is the use of acpcPNA can be an alternative to DNA or RNA probe due to its chemically and biologically stable molecule, easy to be synthesized and hybridized efficiently with the cDNA strands. Besides, the PADs implementation has gained renewed interest from researchers around the world in the fabrication of biosensor due to their several advantages, such as simple, inexpensive, portable and disposable devices, which complement to the characteristics of a POC biosensor [100]. Moreover, a semi-quantitative analysis of colorimetric assays can be accomplished by using the image processor. Despite of the simple technology, the fabrication of the paper-based biosensor needs a long optimization time and requires competencies and expertise [102]. The main advantages and disadvantages of the colorimetric assay using AgNPs colloid are added in section 4.1.3. Correction has been made in page 15, line 30-33, and in page 16, line 1-5 of the revised manuscript.
The main advantage of this work is the application of AuNPs optical properties for effective signal transformation of biological interaction into a physical signal. Therefore, the optical biosensors are selective and sensitive devices for the detection of very low levels of chemicals and biological substances without using electrical fields [105]. Unfortunately, these methods are mostly requiring DNA hybridization label for target detection and take a long assay duration. These strategies are also susceptible to nucleic acid degradation that could render an unexpected error in the actual nucleic acid assay. The main advantages and disadvantages of reflectance using AuNPs-PSA sphere are added in section 4.1.4. Correction has been made in page 16, line 32-38 of the revised manuscript.
The voltammetry-based biosensor, on the other hand, is inherently ultrasensitive towards determination of ultralow nucleic acid concentration at fM levels [106]. The signal amplification from the nanobioconjugates generated a high signal even at an extremely low concentrations of RNA. The high sensitivity of the resulting electrochemical genosensor was attributed to the optimization of several parameters involved, such as enzyme concentration, AuNPs and ssDNA concentrations, hybridization temperature, and spacer/signal molar ratio. Other factors that must also be considered are salt concentration, pH value and temperature, which are responsible for the stability of the nanobioconjugates. This would absolutely be needing extra time, cost and sample handling procedure. The main advantages and disadvantages of voltammetric detection using SPAuE and SPCE/Au are added in section 4.2.1. Correction has been made in page 18, line 43, and in page 19, line 1-8 of the revised manuscript.
The superiority of this homemade biosensor is the rapid detection of less than 5 min of incubation time. Besides, the use of the latest technology integrated with a microcontroller and an algorithm for the computer interface has enabled the real-time recording of
electrical signals, which allowed the sensitive detection of RNA virus SARS-CoV-
2. In addition, there is no need for additional redox medium for electron transfer reaction, fast response time to achieve equilibrium response stage, excellence shelf life, and its plausible economic production. However, the threshold value that was used to determine a positive or negative sample may be subject to change with an increasing number of tested clinical samples. Hence it is crucial for the researchers to set a new threshold value for different number of samples. The main advantages and disadvantage of voltammetric detection using AuNPs capped with highly specific antisense oligonucleotides are added in section 4.2.1. Correction has been made in page 20, line 11-19 of the revised manuscript.
The main benefits of this impedance-based biosensor are cost-effective and simple fabrication of the biosensor as it is a label-free approach in DNA detection. This will avoid the non-specific signal from the fluorescent probes that can be detected during the DNA hybridization process. Hence, the specificity of RNA detection could be increased. This fabrication also enables the detection of oxidation signal from the enzymatic product by DPV technique in the presence of target RNA. Despite of the simple design, an issue that needs to be addressed is the non-specific adsorption during impedance biosensing, which can still be solved by using several solutions [126]. The main advantages and disadvantage of impedimetric detection using AuNPs capped with highly specific antisense oligonucleotides are added in section 4.2.2. Correction has been made in page 21, line 7-14 of the revised manuscript.
- If possible, the authors should distinguish between sensitivity of the different biosensors presented. Do the cited studies address that? Please argue.
ANSWER- The low LOD values of viral RNA detection acquired by using various types of Au and Ag nanoparticles-based biosensors are tabulated in Table 1. Based on the table, all the biosensors demonstrated high sensitivity with a low LOD in the range of fM to nM. The lowest LOD can be observed by the reflectance-based biosensor, which showed the LOD value of 1×10-29 M via sandwich recognition procedure [104]. However, the biosensor was susceptible to nucleic acid degradation that resulting in the unexpected errors in actual nucleic acid detection. Hence, several optimizations and modifications of the biosensors are needed to overcome the problem. Another biosensor which demonstrated the lowest LOD value (0.2 fM) was detected by DPV based on the sandwich hybridization assay [119]. This shows that the sandwich-type biosensor design yields the highest sensitivity, which can afford to detect the target nucleic acid at extremely low concentrations. An explanation to the highest sensitivity in comparison with the LOD values of all the biosensors is added in page 21, line 31-41. However, the cited studies did not much argue on the sensitivity of their biosensor.
- The authors state “the absorption band splits into two parts when the symmetry is reduced from spherical to cylindrical such as nanorods,” However, this statement should be edited, since the optical absorption band in plasmonic nanoparticles always have two components regarding the vectorial nature of light. When symmetry between the minor and the mayor axis in a nanoparticle is present, a coincidence in the Surface Plasmon Resonance bands occurs. Otherwise a shift between the absorption bands can be easily observed by the orthogonal components of polarization of light. You can see for instance: doi:10.1088/2040-8978/14/12/125203. It is suggested to clarify this information within the text.
ANSWER- The statement on “the absorption band splits into two parts when the symmetry is reduced from spherical to cylindrical such as nanorods,” has changed to “two SPR bands can be observed when the symmetry is reduced from spherical to cylindrical such as nanorods,”. The revised statement is in agreement to the statement that the optical absorption band in plasmonic nanoparticles always possessing two components regarding the vectorial nature of light. Correction has been made in page 4, line 40-41.
- A summarizing schematic figure for describing the information in part 4 would be helpful. Maybe a small figure would be nice to summarize the current knowledge.
ANSWER- The schematic figure is added in page 22, in line 1-19.
- The discussion section should present a confrontation of the main results with comparative publication promoting additional solutions in biological sensing by nanoparticles. You can consider for instance nonlinear optic effects or signal modulation. You can see for instance: doi:10.3390/s19214728
ANSWER- In addition, the evolution of the optical absorptive effects exhibited by plasmonic nanoparticles enable the improvement in the biological sensing. One of the approaches is by using chaotic signal circuit, which can exhibit powerful nonlinear optical responses arising from chaotic behavior in the areas of photonics and plasmonics [129]. By using this approach, the small changes in plasmonic signals of the nanoparticles can be recorded with the assistance of near-resonance optical excitations. The additional solution in biological sensing by nanoparticles is added in the conclusion. Correction has been made in page 26, line 8-13.
- Only few references were included in the bibliography for this year 2021. I consider that some references can be updated in order to see the current panoramic research in this field.
ANSWER- An article of LSPR-based biosensor has recently been published in 2021 and the reference is added in the bibiliography as below:
- Qiu, G.; Gai, Z.; Saleh, L.; Tang, J.; Gui, T.; Kullak-Ublick, G.A.; Wang, J. Thermoplasmonic-Assisted Cyclic Cleavage Amplification for Self-Validating Plasmonic Detection of SARS-CoV-2 ACS Nano 2021, 15, 7536-754.
The references (2021) for the applications of gold-silver nanoparticles are added in the bibiliography as below:
- Zhao, K.; Ge, L.; Wong, T.I.; Zhou, X.; Lisak, G. Gold-silver nanoparticles modified electrochemical sensor array for simultaneous determination of chromium(III) and chromium(VI) in wastewater samples. Chemosphere, 2021, 281, 130880.
The references (2021) for the current applications using the similar approaches to the biosensor design are added in the bibiliography as below:
- Jia, Z.; Tu, K.; Xu, Q.; Gao, W.; Liu, C.; Fang, B.; Zhang, M. A novel disease-associated nucleic acid sensing platform based on split DNA-scaffolded sliver nanocluster. Anal. Chim. Acta 2021, 1175, 338734.
- Zhou, Z.; Xiao, R.; Cheng, S.; Wang, S.; Shi, L.; Wang, C.; Qi, K.; Wang, S. A universal SERS-label immunoassay for pathogen bacteria detection based on Fe3O4@Au- aptamer separation and antibody-protein A orientation recognition. Anal. Chim. Acta 2021, 1160, 338421.
The references (2021) which support the advantages and disadvantages of each biosensors and are added bibliography as below:
- Zhang Y.; An, W.; Zhao, C.; Dong, Q. Radiation induced plasmonic nanobubbles: fundamentals, applications and prospects. AIMS Energy, 2021, 9, 676-713.
- Kemp, N.T. A tutorial on electrochemical impedance spectroscopy and nanogap electrodes for biosensing applications. IEEE Sens. J. 2021, doi: 10.1109/JSEN.2021.3084284.
- Collective citations in the style of [41,42] should be individually presented in order to better justify the importance of each citation selected for this review.
ANSWER- The citation for number 42 has removed and only statement from citation number 40 is remained. However, the number is changed to 53 due to new references are added in this revised manuscript. Correction has been made in page 5, line 34.

Reviewer 3 Report
The emergence of highly pathogenic and deadly human coronaviruses have threatened to human health across the world. The early and rapid detection is required in order to provide a suitable treatment for the containment of the diseases. Based on the advance of the nanotechnology, this article reviews the development of biosensor for the detection of RNA virus based on gold and silver nanoparticles. The authors review the properties and ligand choice of AuNPs and AgNPs in detail. However, a comprehensive review on the design of the sensors in each aspect such as fluorescence-based assay, colorimetric assay and LSPR etc. should also be carried out, obviously, there is insufficient literature cited. Therefore, I don not recommend the publication. In addition, there are some issues should be revised.
- The “Figure 2” in Page 9, line 6 should be revised as “Figure 1”.
- The “real-time reverse transcription-polymerase chain reaction” in Page 2 last two lines should be revised as “real-time quantitative reverse transcription-polymerase chain reaction”.
- References in Table 2 and the captions of all Figures and do not correspond to citations in the main text. The caption of Figure 4 contains no reference.
Author Response
Response to Reviewer #3
The emergence of highly pathogenic and deadly human coronaviruses have threatened to human health across the world. The early and rapid detection is required in order to provide a suitable treatment for the containment of the diseases. Based on the advance of the nanotechnology, this article reviews the development of biosensor for the detection of RNA virus based on gold and silver nanoparticles.
- The authors review the properties and ligand choice of AuNPs and AgNPs in detail. However, a comprehensive review on the design of the sensors in each aspect such as fluorescence-based assay, colorimetric assay and LSPR etc. should also be carried out, obviously, there is insufficient literature cited. Therefore, I don not recommend the publication. In addition, there are some issues should be revised.
ANSWER- Based on the several comments from all reviewers, this review has been edited and added with many additional information (for texts that are marked in red) to provide a comprehensive review on the design of gold and silver nanoparticles-based biosensor. The current applications using the similar approaches to the biosensor design are added in the explanation in the main text. This review has also wrapped-up the advantages and disadvantages for each type of biosensor discussed. This might be helpful for other researchers to make decision and consideration for the development of a more efficient, cost-effective and novel biosensor.
- The “Figure 2” in Page 9, line 6 should be revised as “Figure 1”.
ANSWER- The “Figure 2” in the mentioned page and line is maintained as we have added another schematic figure based on another reviewer’s comment for Figure 1.
- The “real-time reverse transcription-polymerase chain reaction” in Page 2 last two lines should be revised as “real-time quantitative reverse transcription-polymerase chain reaction”.
ANSWER- The “real-time reverse transcription-polymerase chain reaction” has changed to “real-time quantitative reverse transcription-polymerase chain reaction”. Correction has been made in page 2, line 41.
- References in Table 2 and the captions of all Figures and do not correspond to citations in the main text. The caption of Figure 4 contains no reference.
ANSWER- There are some technical issues regarding the numbering of citations in the Figures and Table 1. We apologize for the inconveniences caused. However, the citations in Table 1 and the captions of all Figures have been revised corresponding to the citations in the main text. The citation number and reference of the previous Figure 4 has been corrected accordingly.

Reviewer 4 Report
The authors describe the recent developments of biosensors based on metal nanoparticles (MeNPs) for pathogenic RNA viruses’ detection. The properties of MeNPs that are exploited in the construction of optical and electrochemical biosensors are discussed. The detection strategies based on optical and electrochemical transducers are briefly described. The comparison of the analytical performance amongst various biosensors is merely based on detection limit. The work is well organized and the review describes the recent developments in the use of MeNPs in biosensors technology.
The manuscript can be considered for publication after minor revision:
1. Table 1 is not needed as long as the main results are discussed within the text.
2. The functioning principle of the electrochemical biosensors mainly voltammetric and impedimetric ones lacks clarity. The functioning principle should be briefly described.
3. Table 2 displays a limited number of examples. It is recommended that several works published in the last five years shall be included in Table 2 to provide an updated overview on the discussed topic.
4. The references should be checked to ensure that proper citation details are given for accuracy.
Minor points:
1. The text should be checked for typesetting errors and typos.
Author Response
Response to Reviewer #4
The authors describe the recent developments of biosensors based on metal nanoparticles (MeNPs) for pathogenic RNA viruses’ detection. The properties of MeNPs that are exploited in the construction of optical and electrochemical biosensors are discussed. The detection strategies based on optical and electrochemical transducers are briefly described. The comparison of the analytical performance amongst various biosensors is merely based on detection limit. The work is well organized and the review describes the recent developments in the use of MeNPs in biosensors technology. The manuscript can be considered for publication after minor revision:
- Table 1 is not needed as long as the main results are discussed within the text.
ANSWER- Table 1 has been removed as per suggested, and the main results are remained within the text.
- The functioning principle of the electrochemical biosensors mainly voltammetric and impedimetric ones lacks clarity. The functioning principle should be briefly described.
ANSWER- The applied potential is responsible for a change in the concentration of an electroactive species at the working electrode as the redox reaction taking place. When a suitable potential is reached, the analyte current begins to flow [115]. The current is called a cathodic current when a reduction occurs and an anodic current when oxidation occurs. The functioning principle for voltammetric is added in section 4.2.1. Correction has been made in page 18, line 21-24.
For example, [Fe(CN)6]3−/4− in the solution are repelled by the negatively charged species, which can increase the electron transfer resistance. The increment in electron transfer resistance can be measured as a function of analyte concentration by using electrochemical impedance spectroscopy, which provides quantitative parameters of electrochemical processes [114]. The functioning principle for impedimetric detection is added in section 4.2.2. Correction has been made in page 20, line 34-38.
- Table 2 displays a limited number of examples. It is recommended that several works published in the last five years shall be included in Table 2 to provide an updated overview on the discussed topic.
ANSWER- An article of LSPR-based biosensor has recently been published in 2021 and the summary of the biosensor development is added in Table 1 and in section 4.1.2 (in page 12, line 21-31, and in page 13, line 1-28). There are limited number of examples in Table 1 as the currently developed biosensors based on incorporation of Au and Ag for detection of pathogenic RNA viruses are quite limited. However, all the biosensors stated in Table 1 were published in the last five years i.e. from 2017 and onwards except for the only one biosensor that was published in 2015.
- The references should be checked to ensure that proper citation details are given for accuracy.
ANSWER- The references in this review have been properly and thoroughly checked. We apologiz e for the inconveniences caused as there were some technical issues in the previous review.
Minor points:
- The text should be checked for typesetting errors and typos.
ANSWER- Corrections have been done accordingly for texts that are marked in red.

Round 2
Reviewer 1 Report
The authors have satisfactorily addressed the Reviewers' comments and have taken in consideration seriously their suggestions. I have no further comment.
Reviewer 2 Report
The authors have successfully clarified the points raised in the review stage. The results are valuable and well presented. Then, in my opinion, the report can be considered for publication as it is.
Reviewer 3 Report
The authors have revised this review and added the current applications using the similar approaches to the biosensor design, as well as the advantages and disadvantages for each application. After this major revision, this article provide a comprehensive review on the design of gold and silver nanoparticles-based biosensor and can be published in present form.